# REFERENCE-GUIDED POLICY OPTIMIZATION FOR MOLECULAR OPTIMIZATION VIA LLM REASONING

**Xuan Li**[1]    **Zhanke Zhou**[1]    **Zongze Li**[1]    **Jiangchao Yao**[2]    **Yu Rong**[3,4]
**Lu Zhang**[1]    **Bo Han**[1†]

[1]Hong Kong Baptist University    [2]CMIC, Shanghai Jiao Tong University
[3]DAMO Academy, Alibaba Group    [4] Hupan Lab

## ABSTRACT

Large language models (LLMs) benefit substantially from supervised fine-tuning (SFT) and reinforcement learning with verifiable rewards (RLVR) in reasoning tasks. However, these recipes perform poorly in instruction-based molecular optimization, where each data point typically provides only a single optimized reference molecule and no step-by-step optimization trajectory. We reveal that answer-only SFT on the reference molecules collapses reasoning, and RLVR provides sparse feedback under similarity constraints due to the model's lack of effective exploration, which slows learning and limits optimization. To encourage the exploration of new molecules while balancing the exploitation of the reference molecules, we introduce *Reference-guided **P**olicy **O**ptimization* (RePO), an optimization approach that learns from reference molecules without requiring trajectory data. At each update, RePO samples candidate molecules with their intermediate reasoning trajectories from the model and trains the model using verifiable rewards that measure property satisfaction under similarity constraints in an RL manner. Meanwhile, it applies reference guidance by keeping the policy's intermediate reasoning trajectory as context and training only the answer in a supervised manner. Together, the RL term promotes exploration, while the guidance term mitigates reward sparsity and stabilizes training by grounding outputs to references when many valid molecular edits exist. Across molecular optimization benchmarks, RePO consistently outperforms SFT and RLVR baselines (e.g., GRPO), achieving improvements on the optimization metric (Success Rate $\times$ Similarity), improving balance across competing objectives, and generalizing better to unseen instruction styles. Our code is publicly available at https://github.com/tmlr-group/RePO.

## 1 INTRODUCTION

LLMs have advanced rapidly on reasoning-intensive tasks through supervised fine-tuning (SFT) and reinforcement learning with verifiable rewards (RLVR) (Guo et al., 2025; Muennighoff et al., 2025; Zhang et al., 2025b). Both paradigms allow LLMs to perform deliberative reasoning before generating answers and solve problems requiring multiple reasoning steps (Zelikman et al., 2022; Zhou et al., 2024; 2025a). However, how these recipes perform in scientific tasks remains under-investigated.

In this paper, we study *instruction-based molecular optimization*, where the model is required to optimize the molecular property while maintaining structural similarity with the original molecule (Fig. 1). This task involves competing objectives: improving a target property requires non-trivial structural edits (e.g., adding functional groups), but those edits will reduce structural similarity to the original molecule and can even break chemical validity (Liao et al., 2024; López-Pérez et al., 2024).

The data point in most datasets, however, provides only a single optimized molecule for reference, without a reasoning trajectory. This supervision mismatch raises a key challenge: how to support (i) effective exploration of the constrained chemical space for molecules with desired properties and (ii) answer-level anchoring to the available references to exploit dataset-provided reference molecules.

---

[†]Correspondence to Bo Han (bhanml@comp.hkbu.edu.hk).

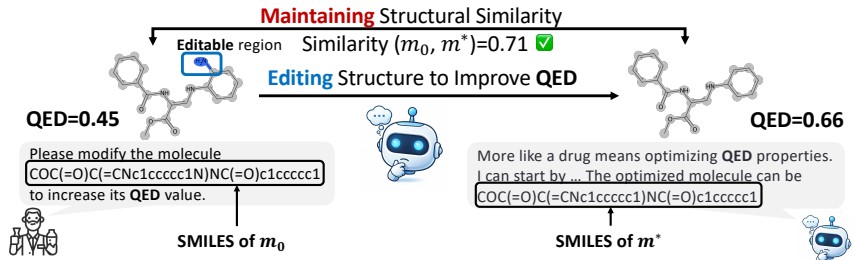

Figure 1: Molecular optimization aims to optimize the given molecule by modifying its components while maintaining the structural similarity of the original molecule after modification. The molecule is presented as SMILES (Weininger, 1988), a sequence of symbols representing atoms and bonds.

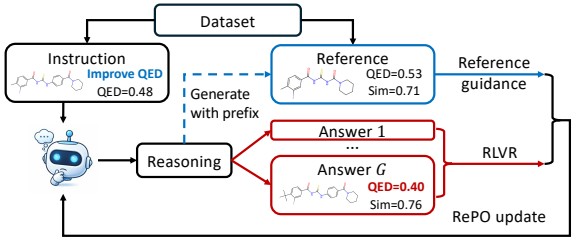

Figure 2: Illustration of RePO. The model generates answers via reasoning; reference guidance anchors to the reference conditioned on the reasoning context, while RLVR optimizes the property under similarity constraints.

Figure 3: Performance comparison on molecular optimization tasks. Details of molecular properties can be found in Appendix I.1.

Notably, established RLVR/SFT recipes face substantial limitations in this domain (Yue et al., 2025; Gandhi et al., 2025; Chen et al., 2025). We show that, with only reference molecules as supervision, SFT often collapses into deterministic, answer-only outputs and suppresses the multi-step reasoning needed for exploration. When RL starts from such an SFT initialization, RLVR updates often fail to produce multi-step reasoning trajectories and instead preserve the short-response style. Conversely, direct RL from the base model depends on rare answers that both improve the property and satisfy similarity; early successes are usually tiny edits with marginal gains, which drive conservative optimizations, yield sparse learning signals, and limit final performance.

To address these limitations, we propose Reference-guided Policy Optimization (RePO), an optimization approach that provides reference guidance while permitting many valid molecular edits (Fig. 2). Concretely, each RePO step samples candidate molecules from the model given an instruction, scores them with a verifiable reward (property satisfaction while enforcing similarity), and updates the policy with three terms: (i) an RLVR term that upweights higher-reward candidates and lowers the low-reward ones, (ii) an answer-level reference-guidance term that increases the likelihood of the reference conditioned on the model's sampled trajectories, and (iii) a KL regularizer that stabilizes updates. Overall, RePO couples reward-driven exploration in the constrained chemical space with answer-level anchoring to the references, without requiring any labeled intermediate trajectories.

We evaluate RePO on instruction-based molecular optimization benchmarks, including TOMG-Bench (Li et al., 2024a) and MuMOInstruct (Dey et al., 2025). On TOMG-Bench single-objective tasks, RePO achieves the best Success Rate × Similarity in four out of six tasks, with up to 17.4% success rate improvement over GRPO (Fig. 3). On MuMOInstruct multi-objective tasks, RePO better balances competing objectives and maintains its advantage under unseen instruction styles. RePO also generalizes across model families and benefits from increased inference-time compute.

We summarize our contributions as follows:

- We reveal a supervision mismatch in instruction-based molecular optimization, which makes answer-only SFT collapse multi-step exploration and limit RLVR's performance (Sec. 3).

- We propose RePO, a reference-guided policy optimization approach that combines GRPO-style reward-driven exploration with reference guidance conditioned on the model's trajectories (Sec. 4).

- We show that RePO improves optimization metrics, generalizes to unseen instruction styles, and benefits from more generation budget during inference (Sec. 5).

## 2 PRELIMINARIES

**Task formulation.** Molecular optimization aims to modify an input molecule to optimize target properties while preserving structural similarity (López-Pérez et al., 2024; Lipinski & Hopkins, 2004). As illustrated in Fig. 1, we denote the query as $q = (x, m_0)$, where $x$ is the textual instruction and $m_0$ is the input molecule. The model outputs a response $o = [t; \hat{m}]$ consisting of reasoning tokens $t$ and an optimized molecule $\hat{m}$. Given the candidate molecule space $\mathcal{M}$, target-property function $F : \mathcal{M} \to \mathbb{R}$, similarity function $\text{Similarity}(\cdot, \cdot)$, and threshold $\delta \in [0, 1]$, the task is formulated as

$$m^* = \arg \max_{m \in \mathcal{M}} F(m) \quad \text{s.t.} \quad \text{Similarity}(m, m_0) \geq \delta. \tag{1}$$

Here $m^*$ denotes an optimized molecule that satisfies both property and similarity constraints. We use $m$ for a generic molecule, $m_0$ for the input molecule in $q$. Each dataset instance also provides a reference molecule, denoted by $m_{\text{ref}}$, which serves as a reference for $m^*$ and is used as the guidance target in Sec. 4. We provide additional discussions in Appendix G.1.

**LLM reasoning in molecular optimization.** Despite their merits, conventional molecular design approaches struggle to synthesize molecules with precisely tailored properties (Li et al., 2024b;c) and often generalize poorly to novel objectives or constraint combinations (Dey et al., 2025; Li et al., 2024a; Zhang et al., 2025a). These limitations motivate us to leverage LLMs' ability to follow natural-language instructions (Chang et al., 2024) for optimizing molecules. In addition, prior work suggests LLMs can capture non-trivial knowledge of molecular properties (Guo et al., 2023).

Nevertheless, LLMs still struggle under chemistry-specific constraints. Benchmarks such as SciBench (Wang et al., 2023a) and ChemBench (Mirza et al., 2025) report substantial degradation when tasks require validity preservation or property-preference satisfaction. Accordingly, recent GPT-based molecular optimization methods often embed LLMs in black-box pipelines: MOLLEO (Wang et al., 2025b) uses evolutionary search with LLM-proposed edits, while others fine-tune LLMs as scoring oracles (Nguyen & Grover, 2025) or couple them with chemistry tools (Bran et al., 2023).

**RLVR for LLM reasoning.** We adopt GRPO (Shao et al., 2024) for training the LLM in the RLVR paradigm. For data pair $(q, m_{\text{ref}}) \sim \mathcal{D}$, the old policy $\pi_{\text{old}}$ samples $G$ responses $\{o_i\}_{i=1}^{G}$. Let $r(o_i, q)$ be the scalar reward computed from the molecule $m_i$ in $o_i$ and the input molecule $m_0$ in $q$, and $\pi_{\text{ref}}$ be the reference policy for KL regularization. GRPO maximizes the following objective

$$\mathcal{J}_{\text{GRPO}}(\pi_\theta) = \mathbb{E}_{(q, m_{\text{ref}}) \sim \mathcal{D}, \{o_i\}_{i=1}^{G} \sim \pi_{\text{old}}(\cdot|q)} \left[ \frac{1}{G} \sum_{i=1}^{G} \frac{1}{|o_i|} \sum_{k=1}^{|o_i|} \left( \min \left( \frac{\pi_\theta(o_{i,k}|q, o_{i,<k})}{\pi_{\text{old}}(o_{i,k}|q, o_{i,<k})} \hat{A}_{i,k}, \right. \right. \right.$$
$$\left. \left. \left. \text{clip} \left( \frac{\pi_\theta(o_{i,k}|q, o_{i,<k})}{\pi_{\text{old}}(o_{i,k}|q, o_{i,<k})}, 1 - \varepsilon, 1 + \varepsilon \right) \hat{A}_{i,k} \right) - \gamma \, \mathbb{D}_{\text{KL}}(\pi_\theta \| \pi_{\text{ref}}) \right) \right], \tag{2}$$

where $\hat{A}_{i,k} = (r(o_i, q) - \text{mean}(\{r(o_j, q)\}_{j=1}^{G})) / \text{std}(\{r(o_j, q)\}_{j=1}^{G})$ is the token-wise group-relative advantage. We use the K3 estimator (Schulman., 2020) for calculating the KL regularization:

$$\mathbb{D}_{\text{KL}}(\pi_\theta \| \pi_{\text{ref}}) = \frac{\pi_{\text{ref}}(o_{i,k}|q, o_{i,<k})}{\pi_\theta(o_{i,k}|q, o_{i,<k})} - \log \frac{\pi_{\text{ref}}(o_{i,k}|q, o_{i,<k})}{\pi_\theta(o_{i,k}|q, o_{i,<k})} - 1. \tag{3}$$

## 3 SUPERVISION MISMATCH UNDER COMPETING OBJECTIVES

Eqn. 2 improves the policy by increasing the likelihood of responses that achieve higher reward. In instruction-based molecular optimization, however, high-reward samples are often rare early in training: among generated molecules, only a small fraction both (i) achieves meaningful target-property improvement and (ii) satisfies the similarity constraint. In addition, each data point typically provides only a single reference molecule $m_{\text{ref}}$ without intermediate edit trajectories. This supervision mismatch leaves the model with little guidance on how to transform $m_0$ into better molecules under the constraint, causing the policy to be conservative, near-identity edits.

To examine how these issues manifest in practice, we compare three training recipes under matched settings: (i) GRPO from a base model, (ii) final-answer-only SFT from a base model, and (iii) GRPO initialized from a final-answer-only SFT model, denoted GRPO (SFT-init). We report success rate,

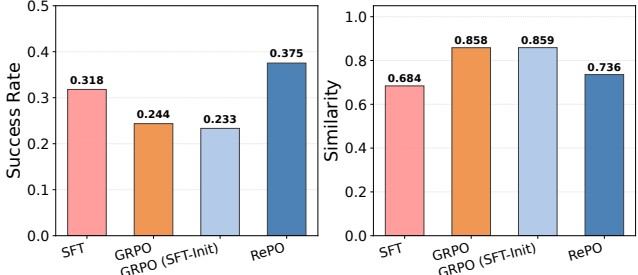 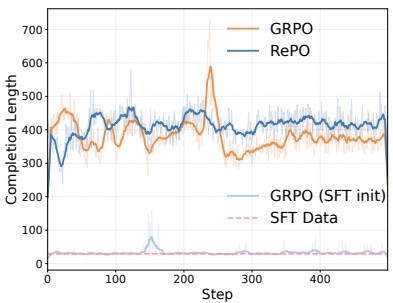

Figure 4: Average success rate and similarity for SFT, GRPO, GRPO (SFT-init), and RePO on property optimization. GRPO achieves high similarity but a low success rate. RePO improves success while maintaining high similarity.

Figure 5: Training dynamics of completion lengths of different methods. GRPO (SFT-init) generates short responses that match the SFT data.

similarity on the test set, and response length during training (as a proxy for whether multi-step reasoning is expressed). We employ `Qwen-2.5-3B Instruct` (Qwen et al., 2024) as the base model and the property optimization task of TOMG-Bench for training and evaluation. We employ structural similarity (Eqn. 5) and target properties (Eqn. 6) as rewards for GRPO training. Detailed settings are provided in Appendix I. Figs. 4 and 5 summarize the results, leading to three observations.

**Observation 3.1** (*GRPO on the base model struggles under competing objectives in molecular optimization*). GRPO tends to generate *conservative edits* that remain very close to the input molecule $m_0$. These edits keep similarity high, but yield limited property improvement (Fig. 4). As a result, most sampled molecules contribute weak learning signals, and updates concentrate on a narrow local regime rather than exploring larger yet promising edits in chemical space.

**Observation 3.2** (*SFT on the base model produces answer-only outputs and weak similarity control*). Answer-only SFT trains the model to directly imitate the reference molecule, producing short, answer-only outputs whose lengths match the SFT training data (dashed line in Fig. 5). While SFT attains a moderate success rate, it yields lower similarity than RL methods (Fig. 4), indicating that token-level imitation of the reference alone provides limited control over the similarity-property trade-off.

Since SFT collapses outputs into short, answer-only responses, we next ask whether applying GRPO on top of this SFT initialization can recover multi-step reasoning.

**Observation 3.3** (*GRPO on the SFT model does not recover multi-step reasoning*). In Fig. 5, GRPO (SFT-init) largely preserves the short-response style, producing completion lengths that closely match the SFT data rather than recovering the multi-step reasoning observed in GRPO from the base model. This short-response behavior carries over to test performance: GRPO (SFT-init) achieves the lowest success rate among all methods while maintaining high similarity (Fig. 4), indicating that it inherits the SFT model's limited exploration without gaining RL-driven property improvement.

Taken together, these observations highlight a modeling gap: reward-only updates are often too sparse to reliably push the policy beyond conservative edits, yet naive token-level imitation of the reference molecule can over-constrain the policy and suppress diverse edit trajectories. An improved approach should therefore provide (i) directional, answer-level reference guidance toward $m_{\text{ref}}$ enhancing the learning signal, while (ii) avoiding token-level process imitation so that multiple valid reasoning traces and edit paths remain possible under the same instruction.

## 4 RePO: REFERENCE-GUIDED POLICY OPTIMIZATION

The previous analysis identifies two desiderata: answer-level guidance that enhances learning signal and trajectory-level RL updates that avoid token-level imitation and preserve diverse exploration. RePO satisfies both by using the reference molecule as an *answer-level anchor*, while retaining RLVR-style updates on the full reasoning trajectories to preserve exploration over diverse edits.

**Objective.** For each query $q = (x, m_0)$, we sample $\{o_i\}_{i=1}^{G} \sim \pi_{\text{old}}(\cdot \mid q)$, where each response is partitioned as $o_i = [t_i; \hat{m}_i]$ with reasoning tokens $t_i$ and a final molecule $\hat{m}_i$ (Fig. 6). Let $r(o_i, q)$ be the scalar reward computed from $\hat{m}_i$ and the input molecule $m_0$ in $q$ (Sec. 4). We compute the

token-wise group-relative advantage $\hat{A}_{i,k}$ as in Eqn. 2. The RePO objective is

$$\mathcal{J}_{\text{RePO}}(\pi_\theta) = \mathbb{E}_{\substack{(q,m_{\text{ref}})\sim\mathcal{D} \\ \{o_i\}_{i=1}^G\sim\pi_{\text{old}}(\cdot|q)}} \left[ \frac{1}{G}\sum_{i=1}^G \Big( \underbrace{\frac{1}{|o_i|}\sum_{k=1}^{|o_i|}\min\Big(\rho_{i,k}\hat{A}_{i,k},\text{clip}(\rho_{i,k},1-\varepsilon,1+\varepsilon)\hat{A}_{i,k}\Big)}_{\text{Exploration term}} \right.$$

$$\left. + \beta\underbrace{\log\pi_\theta(m_{\text{ref}}\mid q,t_i)}_{\text{Answer-level reference guidance}} \underbrace{-\gamma\frac{1}{|o_i|}\sum_{k=1}^{|o_i|}\mathbb{D}_{\text{KL}}(\pi_\theta\|\pi_{\text{ref}})}_{\text{KL regularization}} \Big) \right], \qquad (4)$$

where $\rho_{i,k} = \pi_\theta(o_{i,k}\mid q,o_{i,<k})/\pi_{\text{old}}(o_{i,k}\mid q,o_{i,<k})$, and $\mathbb{D}_{\text{KL}}(\pi_\theta\|\pi_{\text{ref}})$ uses the estimator as Eqn. 2. The exploration term applies updates to all tokens in $o_i$. The guidance term increases the likelihood of the reference molecule conditioned on the reasoning prefix $t_i$, and $\beta$ controls its strength.

We next provide a detailed description of the remaining key design choices in our framework, including (i) the reference molecule, (ii) the reward function and (iii) the overall training procedure.

**Reference molecules.** For each query $q$, we have the dataset providing a reference molecule $m_{\text{ref}}$, which serves as a proxy for the (unknown) optimal solution $m^*$ in Eqn. 1. RePO uses $m_{\text{ref}}$ only as an *answer-level* anchor in Eqn. 4. In practice, we apply RDKit validity checks to $m_{\text{ref}}$; For notation convenience, we continue to denote the checked reference as $m_{\text{ref}}$. Importantly, $m_{\text{ref}}$ does not provide reasoning supervision: RePO does not treat it as a preferred reasoning path, and trajectory learning is driven by rewards on model-sampled responses. We showcase a $m_{\text{ref}}$ as follows:

> **Query:** "Please modify the molecule `CCCC(C(=O)OC)C(O)C1CCCCC1` to increase its LogP value."
> **Reference molecule:** `O=C(OC(CCCO)CCCS)c1ccccc1`

**Reward design.** The reward instantiates the objective in Eqn. 1. For any molecule $m$, let $\text{FP}(m)$ denote its molecular fingerprint, a fixed-length binary vector whose entries indicate the presence of specific substructures in $m$. We define the scalar reward as $r(m,m_0) = r_{\text{prop}}(m,m_0)+r_{\text{struct}}(m,m_0)$.

- **Structural similarity** $r_{\text{struct}}$: we use Tanimoto similarity (Bajusz et al., 2015)

$$r_{\text{struct}}(m,m_0) = \frac{|\text{FP}(m)\cap\text{FP}(m_0)|}{|\text{FP}(m)\cup\text{FP}(m_0)|} \in [0,1]. \qquad (5)$$

This term encourages structural preservation by assigning higher rewards to molecules close to $m_0$.

- **Target property** $r_{\text{prop}}$: the instruction specifies a target property and whether it should be increased or decreased (e.g., increasing the molecular LogP value); let $F(\cdot)$ denote the corresponding property function. We use a binary improvement property reward defined as follows:

$$r_{\text{prop}}(m,m_0) = \begin{cases} 1, & \text{if } F(m)\geq F(m_0) \text{ (for increasing)}, \\ 1, & \text{if } F(m)\leq F(m_0) \text{ (for decreasing)}, \\ 0, & \text{otherwise}. \end{cases} \qquad (6)$$

**Training.** For each query $q$, we sample $\{o_i = [t_i;\hat{m}_i]\}_{i=1}^G$ and compute rewards $r(o_i,q)$ on the parsed $\hat{m}_i$ (with $m_0$ from $q$). We then (i) compute group-relative advantages and apply the clipped GRPO update to *all tokens* in $o_i$ (the GRPO term in Eqn. 4); and (ii) add the reference-guidance term $\log\pi_\theta(m_{\text{ref}}\mid q,t_i)$, which is computed over the *answer tokens* only, with the sampled reasoning prefix $t_i$ as context (Fig. 6). Thus, RePO does not imitate reasoning tokens; instead, it leverages $m_{\text{ref}}$ to provide an answer-level anchor while letting RL update on sampled trajectories.

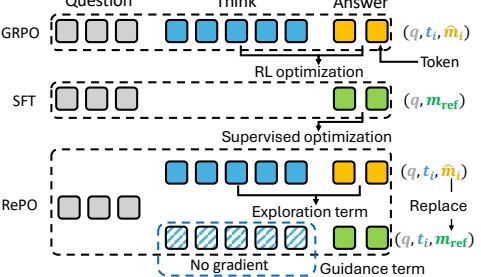

Figure 6: Token partition and gradient application for SFT, GRPO, and RePO.

**Remark 4.1.** Reference guidance operates solely at the answer level: it increases the likelihood of instruction-satisfying answers early in training, reducing reward sparsity and yielding more informative RL updates, which in turn shape the reasoning tokens and support model to explore molecules with better properties while satisfying the constraint.

Table 1: SuccessRate (SR), Similarity (Sim), and their product (SR×Sim) on TOMG-Bench for structure- and property-based optimization tasks. The higher the better. The best results for each task are **bolded**, and the second-best results are underlined.

| Task type | Objective | Metric | Base Model | Distill-SFT | SFT | GRPO | GRPO (SFT init) | RePO |
|---|---|---|---|---|---|---|---|---|
| Structure-based optimization | AddComponent | SR | 0.086 | 0.100 | 0.238 | 0.005 | 0.246 | **0.307** |
| | | Sim | 0.763 | 0.604 | 0.619 | **0.992** | 0.635 | 0.778 |
| | | SR×Sim | 0.066 | 0.060 | 0.147 | 0.005 | 0.156 | **0.239** |
| | DelComponent | SR | 0.107 | 0.188 | 0.203 | 0.008 | **0.232** | 0.158 |
| | | Sim | 0.864 | 0.682 | 0.755 | **0.994** | 0.759 | 0.887 |
| | | SR×Sim | 0.092 | 0.128 | 0.153 | 0.008 | **0.176** | 0.140 |
| | SubComponent | SR | 0.057 | 0.078 | 0.366 | 0.053 | 0.420 | **0.429** |
| | | Sim | 0.815 | 0.633 | 0.721 | **0.972** | 0.713 | 0.802 |
| | | SR×Sim | 0.046 | 0.049 | 0.264 | 0.052 | 0.299 | **0.344** |
| Property optimization | QED | SR | 0.188 | 0.208 | 0.297 | 0.138 | 0.223 | **0.312** |
| | | Sim | 0.693 | 0.594 | 0.697 | **0.889** | 0.863 | 0.756 |
| | | SR×Sim | 0.130 | 0.124 | 0.207 | 0.123 | 0.192 | **0.236** |
| | LogP | SR | 0.268 | 0.234 | 0.298 | 0.379 | 0.212 | **0.415** |
| | | Sim | 0.627 | 0.579 | 0.692 | 0.806 | **0.863** | 0.715 |
| | | SR×Sim | 0.168 | 0.135 | 0.206 | **0.305** | 0.183 | 0.297 |
| | MR | SR | 0.252 | 0.214 | 0.359 | 0.214 | 0.265 | **0.399** |
| | | Sim | 0.685 | 0.619 | 0.663 | **0.880** | 0.850 | 0.736 |
| | | SR×Sim | 0.173 | 0.132 | 0.238 | 0.188 | 0.225 | **0.294** |

## 5 EXPERIMENTS

We evaluate RePO on instruction-based molecular optimization. For readability, we organize this section into four blocks: setup (Sec. 5.1), main quantitative results (Sec. 5.2), mechanism and robustness analyses (Sec. 5.3), and qualitative plus inference-scaling analyses (Sec. 5.4).

### 5.1 EXPERIMENT SETTINGS

In what follows, we describe the setting of the experiments, including the dataset, baselines, and evaluation metrics. Detailed settings are provided in Appendix I.

**Evaluation Metrics.** We evaluate model performance using three complementary metrics. `Success Rate` is the fraction of test molecules for which the model meets the specified property objective. `Similarity` is measured by the Tanimoto coefficient (Bajusz et al., 2015), which assesses the structural similarity between the input and optimized molecules. Following Li et al. (2024a), to jointly capture both optimization effectiveness and structural preservation, we report `Success Rate × Similarity`, which reflects the model's ability to balance property improvement with maintenance of molecular structure. Detailed discussions on metrics are in Appendix I.1.

**Datasets.** We employ two instruction-based molecular optimization benchmarks, TOMG-Bench (Li et al., 2024a) and MuMOInstruct (Dey et al., 2025), to evaluate the knowledge of LLM on molecular structure and properties. We employ the reference molecules from the molecule optimization datasets as demonstration molecules. We provide a detailed discussion on the validity of the demonstration molecules in Appendix J.5. More detailed description of the datasets in Appendix I.2.

**Baselines.** We use `Qwen-2.5-3B Instruct` as the base model. We compare with Distill-SFT, which applies SFT on the s1.1K dataset (Muennighoff et al., 2025); SFT, which fine-tunes the model on each benchmark training split; GRPO, which applies Eqn. 2; and GRPO (SFT init).

### 5.2 QUANTITATIVE RESULTS

We summarize the main findings from Tabs. 1 and 2.

**RePO elicits the model's chemical reasoning on single-objective optimization tasks.** Tab. 1 summarizes the results for single-objective molecular optimization. For structure-based tasks, RePO achieves the best performance on `AddComponent` and `SubComponent`, corresponding to improvements of 8.3% and 4% over the next best method, respectively. For property-based optimization,

Table 2: Success Rate (SR), Similarity (Sim), and their product (SR×Sim) on MuMOInstruct for seen and unseen instructions. The best results are **bolded**, and the second bests are underlined.

| Task type | Objective | Metric | Base Model | Distill-SFT | SFT | GRPO | GRPO (SFT init) | RePO |
|---|---|---|---|---|---|---|---|---|
| Seen instruction | BDP | SR | 0.052 | 0.078 | **0.398** | 0.156 | 0.088 | 0.206 |
| | | Sim | 0.149 | 0.207 | 0.254 | **0.759** | 0.141 | 0.569 |
| | | SR×Sim | 0.008 | 0.016 | 0.101 | **0.118** | 0.012 | 0.117 |
| | BDQ | SR | 0.034 | 0.022 | **0.319** | 0.082 | 0.022 | 0.160 |
| | | Sim | 0.117 | 0.106 | 0.279 | **0.479** | 0.045 | 0.365 |
| | | SR×Sim | 0.004 | 0.002 | **0.089** | 0.039 | 0.001 | 0.058 |
| | BPQ | SR | 0.052 | 0.064 | **0.471** | 0.212 | 0.056 | 0.274 |
| | | Sim | 0.194 | 0.165 | 0.244 | **0.567** | 0.085 | 0.509 |
| | | SR×Sim | 0.010 | 0.011 | 0.115 | 0.120 | 0.005 | **0.139** |
| Unseen instruction | BDP | SR | 0.052 | 0.016 | **0.310** | 0.148 | 0.092 | 0.198 |
| | | Sim | 0.143 | 0.099 | 0.261 | **0.727** | 0.147 | 0.572 |
| | | SR×Sim | 0.007 | 0.002 | 0.081 | 0.108 | 0.014 | **0.113** |
| | BDQ | SR | 0.042 | 0.020 | **0.342** | 0.078 | 0.026 | 0.170 |
| | | Sim | 0.104 | 0.077 | 0.257 | **0.457** | 0.058 | 0.322 |
| | | SR×Sim | 0.004 | 0.002 | **0.088** | 0.036 | 0.002 | 0.055 |
| | BPQ | SR | 0.050 | 0.050 | **0.419** | 0.186 | 0.042 | 0.242 |
| | | Sim | 0.130 | 0.143 | 0.248 | 0.573 | 0.063 | **0.596** |
| | | SR×Sim | 0.007 | 0.007 | 0.104 | 0.107 | 0.003 | **0.144** |

RePO achieves superior or competitive performance compared to all baselines, highlighting its effectiveness and robustness across evaluation settings, achieving up to 13.0% absolute improvement over the base model. Notably, GRPO without SFT initialization performs markedly worse, particularly on structure-based tasks, underscoring the challenges of unconstrained exploration in the vast chemical space. In contrast, RePO, which integrates demonstration guidance, consistently outperforms SFT and GRPO, yielding more effective molecular optimization.

We also provide experiments with domain-specific optimizers (Bio-T5 (Pei et al., 2023), Mol-T5 (Edwards et al., 2022)) and general LLM-based optimizers (GPT-4o-mini) and multi-round evolutionary methods (Graph-GA (Jensen, 2019), REINVENT (Olivecrona et al., 2017), and MOLLEO (Wang et al., 2025b)) in Tab. 7. Notably, RePO is competitive with or outperforms these methods, despite its single-round optimization and smaller open backbone.

**RePO helps the model to balance multi-objective optimization problems.** Tab. 2 presents the results for multi-objective molecular optimization. Notably, RePO outperforms baseline methods on BDP and BPQ tasks, achieving up to 4% improvements over baseline methods, highlighting its ability to effectively balance multiple competing objectives simultaneously.

**RePO elicit model's generalization ability on unseen instruction styles.** Shown in Tab. 2, the performance advantage of RePO is maintained for unseen instructions, achieving superior results despite the model encountering novel instruction formats. The most significant gains are observed in the BDP task, where RePO's approach to guided exploration proves particularly effective at navigating the complex optimization landscape involving multiple constraints. These results collectively validate that RePO's demonstration-guided approach constrains the exploration space while maintaining the model's reasoning capabilities across scenarios of multi-objective optimization.

## 5.3 MECHANISM AND ROBUSTNESS ANALYSES

We first validate the core mechanism via gradient masking, then analyze training dynamics and outcome distributions, and finally report robustness checks (reasoning-quality evaluation, demonstration quality, prompting baselines, and stronger backbones).

**Mechanism validation via gradient masking.** We ablate gradient masking on TOMG-Bench using RePO with random masking (40%/80%) and a no-mask variant. Fig. 7 shows that all ablations underperform full RePO, and removing masking can even fall below the baseline. This is consistent with the role of masking: without it, the answer-level supervision can backpropagate through intermediate tokens $t_i$, reinforcing spurious or chemically unsound reasoning patterns.

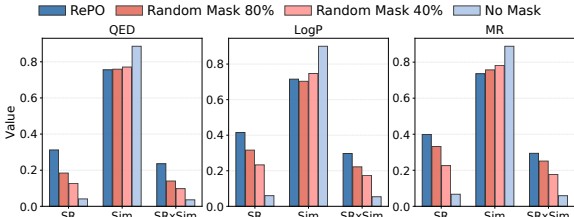

Figure 7: SR, Sim, and SR×Sim for RePO masking ablations on the property optimization task.

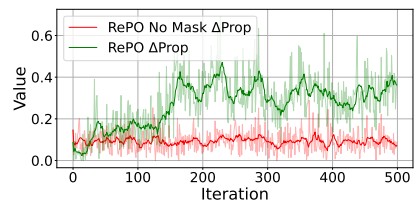

Figure 8: Training reward dynamics for RePO with masking and no-mask variants.

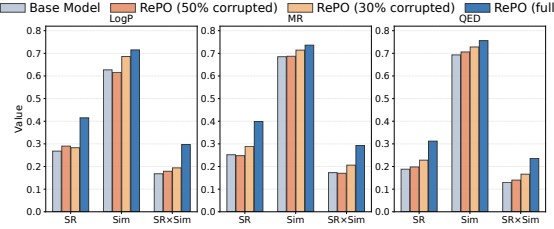

Figure 9: Performance of the base model and RePO corruption variants on property optimization.

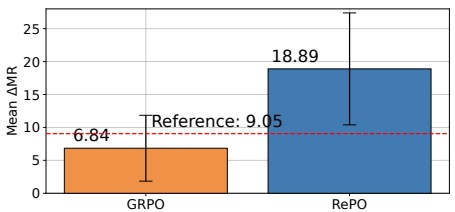

Figure 10: Performance of GRPO, RePO, and reference on MR optimization.

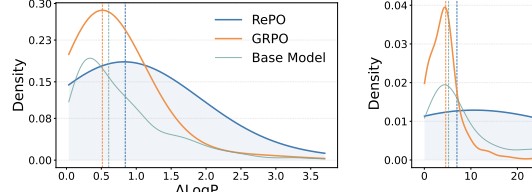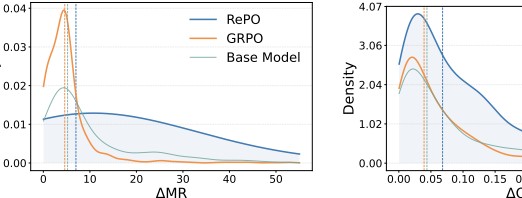

Figure 11: Distribution of property gains on TOMG-Bench property-optimization tasks.

**Masking improves optimization stability.** Fig. 8 compares RePO with and without gradient masking. With masking, the property-improvement reward increases over training; without masking, learning largely stagnates. This trend matches the ablation results in Fig. 7.

**Robustness under reference corruption.** We evaluate robustness by corrupting query-reference alignments within subtasks. As shown in Fig. 9, RePO remains above the baseline with 30% corruption and stays competitive at 50% corruption, indicating graceful degradation under corruption.

**Reference-quality robustness.** We report the full reference-quality ablation in Fig 10. The results show that even reduced reference supervision remains competitive with the GRPO, while full RePO achieves the strongest performance.

**Property-gain distributions show stronger optimization outcomes.** Fig. 11 compares histograms of property gains. RePO shifts toward larger positive gains relative to the base model and GRPO, suggesting that improvements are widespread across candidates rather than driven by a few outliers.

**LLM-as-a-judge evaluation supports reasoning-quality gains.** We evaluate explanation quality with an LLM-as-a-judge protocol calibrated to expert chemists (Zhuang et al., 2025). For TOMG-Bench LogP, we sample 50 trajectories per method and score the associated explanations. Fig. 12 shows that RePO achieves the highest reasoning-quality score.

**RePO also improves larger backbones and different architectures.** We conduct an additional experiment using a larger model, `Qwen-2.5-7B Instruct`. Tab. 4 shows that with a larger model, RePO exceeds the base model by up to 4.9% absolute improvement. In the LogP and MR tasks, RePO also outperforms SFT by a large margin. We also adopt `Llama-3.1-8B-Instruct`, which differs substantially in architecture and tokenizer, and train the model using the TOMG-Bench property optimization task. As shown in Tab. 3, RePO achieves the overall best performance.

**Prompting-only CoT baselines are insufficient.** We compare RePO with zero-shot and few-shot CoT prompting on TOMG-Bench property optimization. As shown in Fig. 13, CoT prompting does

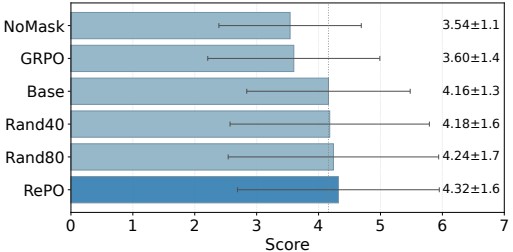

Figure 12: Quantification analysis of the reasoning trajectories' quality.

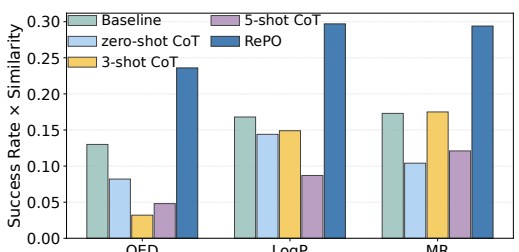

Figure 13: Prompt-based CoT baselines on property optimization.

Table 3: Performance comparison using Llama-3.1-8B Instruct on TOMG-Bench property optimization tasks. Best per row is bolded; second-best is underlined.

|  | Metric | Base Model | SFT | GRPO | GRPO (SFT init) | RePO |
|---|---|---|---|---|---|---|
| LogP | SR | 0.237 | 0.287 | 0.197 | 0.192 | **0.360** |
| | Sim | 0.692 | 0.763 | 0.763 | **0.775** | 0.746 |
| | SR×Sim | 0.164 | 0.219 | 0.151 | 0.149 | **0.269** |
| QED | SR | 0.159 | 0.185 | 0.115 | 0.121 | **0.243** |
| | Sim | 0.722 | 0.809 | 0.805 | **0.811** | 0.783 |
| | SR×Sim | 0.115 | 0.150 | 0.093 | 0.098 | **0.190** |
| MR | SR | 0.169 | 0.230 | 0.142 | 0.141 | **0.293** |
| | Sim | 0.763 | 0.808 | 0.819 | **0.827** | 0.789 |
| | SR×Sim | 0.129 | 0.186 | 0.117 | 0.116 | **0.231** |

Table 4: Performance comparison of the `Qwen-2.5-7B Instruct`.

|  | Baseline | SFT | GRPO | GRPO (SFT init) | RePO |
|---|---|---|---|---|---|
| QED | 0.174 | **0.252** | 0.165 | 0.147 | 0.213 |
| LogP | 0.277 | 0.288 | 0.285 | 0.302 | **0.326** |
| MR | 0.279 | 0.318 | 0.270 | 0.286 | **0.328** |

Table 5: Comparison of RePO with binary and continuous reward functions.

|  | LogP | QED | MR |
|---|---|---|---|
| RePO (binary $r_{\text{prop}}$) | 0.297 | **0.236** | **0.294** |
| RePO (continuous $r_{\text{prop}}$) | **0.301** | 0.203 | 0.292 |

not close the gap and is often worse than the baseline, while RePO consistently performs best. This result indicates that end-to-end policy optimization with reference guidance is more effective than prompt-only reasoning elicitation in this domain.

**Performance of continuous reward.** The property reward $r_{\text{prop}}$ is defined as the improvement in the target property, $\pm(F(m^*) - F(m_0))$, where the sign depends on the optimization direction. In Tab. 5, we report RePO on the property-optimization task of TOMG-Bench. The binary $r_{\text{prop}}$ variant yields more stable gains than the continuous version, suggesting that a discretized success signal is easier to optimize under the similarity constraint.

**Experiments on reward weighting.** Fig. 14 studies sensitivity to the guidance weight $\beta$ on optimizing LogP. $\beta = 0$ reduces to GRPO, while large $\beta$ increasingly resembles SFT. Performance improves as $\beta$ increases, but saturates for $\beta \geq 10$, suggesting that overly strong guidance can limit exploration.

**Out-of-distribution generalization on unseen structure tasks.** We train on TOMG-Bench property optimization and evaluate on withheld structure-editing tasks. Fig. 15 shows that RePO generalizes better than its counterparts, indicating that its learned strategy transfers across instruction types.

## 5.4 CASE STUDIES

**Chemically-validated reasoning.** Fig. 16 illustrates the qualitative differences in reasoning approaches between RePO and GRPO on a molecular optimization task. The left panel demonstrates RePO's chemically sound reasoning process: it correctly identifies structural elements and proposes an effective modification (substituting `Br` with `Cl`). In contrast, GRPO exhibits invalid chemical expressions and proposes chemically implausible modifications (removing nitrogen from a heterocyclic

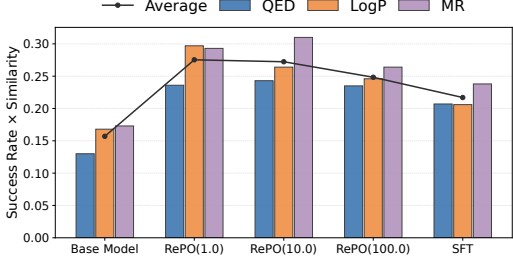

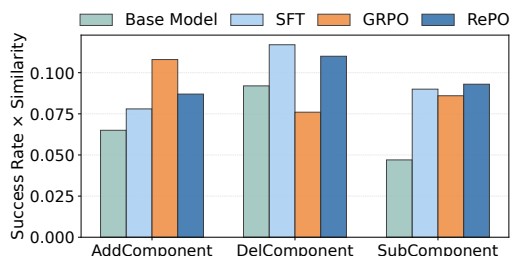

Figure 14: Performance comparison of the base model, RePO with different $\beta$, and SFT.

Figure 15: Performance comparison on unseen structure tasks.

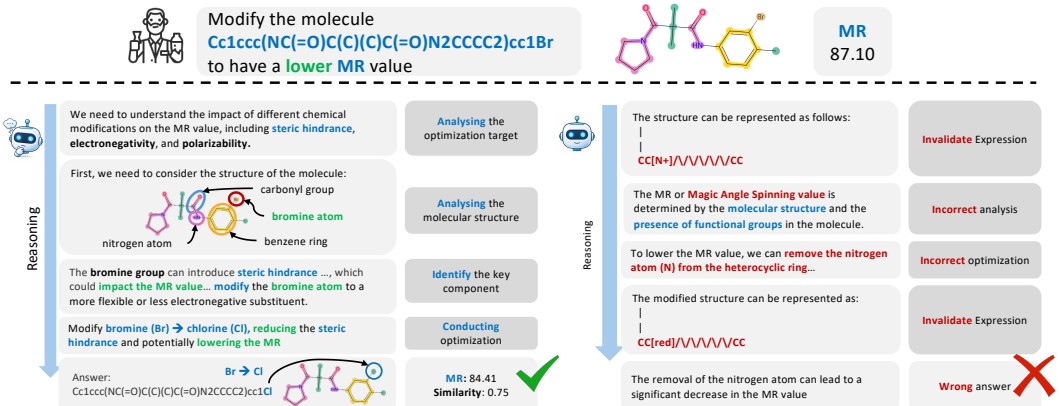

Figure 16: Comparison of RePO (left) and GRPO (right). Notably, RePO employs chemically sound reasoning with valid substitutions, whereas GRPO yields incorrect reasoning and modifications.

ring). This comparison underscores RePO's capacity to generate not only structurally valid molecules but also to produce coherent reasoning that captures the underlying chemical principles governing validated and robust molecular property optimization.

**Inference-scaling properties.** Fig. 17 shows RePO's inference-scaling characteristics. We sample multiple times from the subset of the MR optimization task. The plot reveals that as the number of sampling trials ($k$) increases, RePO's best-of-k success rate (solid curve) and the similarity of the trials (dashed curve) both demonstrate marked improvements. These results underscore RePO's proficiency in leveraging increased computational budgets at inference.

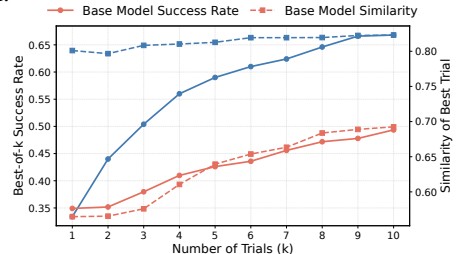

Figure 17: Inference-scaling effect on the success rate and similarity metrics.

## 6 CONCLUSION

We studied the supervision mismatch that arises when applying SFT and RLVR to instruction-based molecular optimization. Our analysis showed that answer-only SFT collapses multi-step reasoning, while reward-only RLVR (e.g., GRPO) under similarity constraints suffers from sparse feedback and conservative optimization. To address this mismatch, we proposed RePO, which balances exploration of new molecules via reward-driven RL updates on sampled trajectories with exploitation of the reference molecules via answer-level reference guidance, regularized by a KL term to stabilize training, all without requiring trajectory supervision. Across TOMG-Bench and MuMOInstruct, RePO consistently outperforms SFT and GRPO baselines on single-objective, multi-objective, unseen-instruction, and inference-scaling evaluations. Future works include improving automatic reference construction and extending to broader scientific optimization domains.

ACKNOWLEDGEMENT

XL, ZKZ, ZZL, and BH were supported by Guangdong Basic and Applied Basic Research Foundation Nos. 2024A1515012399, NSFC General Program No. 62376235, and HKBU CSD Departmental Incentive Scheme. JCY was supported by National Natural Science Foundation of China (No. 62306178) and STCSM (No. 22DZ2229005).

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

APPENDIX

## A    ETHIC STATEMENT

The study does not involve human subjects, data set releases, potentially harmful insights, applications, conflicts of interest, sponsorship, discrimination, bias, fairness concerns, privacy or security issues, legal compliance issues, or research integrity issues.

## B    REPRODUCTION STATEMENT

The experimental setups for training and evaluation are described in detail in Appendix I, and the experiments are all conducted using public datasets. We provide the link to our source codes to ensure the reproducibility of our experimental results: https://github.com/tmlr-group/RePO.

## C    LLM USAGE DISCLOSURE

This submission was prepared with the assistance of LLMs, which were utilized for refining content and checking grammar. The authors assume full responsibility for the entire content of the manuscript, including any potential issues related to plagiarism and factual accuracy. It is confirmed that no LLM is listed as an author.

## D    IMPACT STATEMENT

This paper introduces RePO, a novel framework for enhancing large language model (LLM) reasoning in molecular optimization. By leveraging reference-guided policy optimization, our work aims to accelerate the discovery and design of new molecules, which could have positive impacts in fields such as medicine, materials science, and sustainable chemistry.

## E    LIMITATIONS

Despite RePO's promising results, limitations remain. First, the framework relies on the availability of demonstrations, which may be scarce for novel or complex molecular optimization tasks. In addition, while our approach improves LLM reasoning for molecular optimization, the black-box nature of LLM still presents challenges for domain experts seeking to understand the precise reasoning behind specific structural modifications.

## F    FUTURE WORK

Although RePO is designed to be domain-agnostic, we believe it is particularly well-suited for complex scientific reasoning tasks that, like molecular optimization, involve vast search spaces and require domain-specific, multi-step reasoning—where the solution is difficult to specify but straightforward to verify. We outline two promising directions for future work:

- **Retrosynthesis Planning:** This task aims to predict a sequence of chemical reactions to synthesize a target molecule. The combinatorial explosion of possible reaction pathways makes the search space extremely large, and LLMs may generate chemically invalid or suboptimal steps. By leveraging known synthesis routes from chemical literature as demonstrations, the policy model can be trained to generate retrosynthetic reasoning chains (i.e., stepwise breakdowns of the target molecule). The RL reward can be based on the chemical validity of each proposed reaction (using tools such as ASKCOS (Tu et al., 2025)).

- **Drug-Drug Interaction (DDI) Prediction:** This task involves predicting whether two drugs will interact and providing a mechanistic explanation. Accurate prediction requires understanding complex pharmacological mechanisms (e.g., metabolic pathways), and LLMs are prone to generating plausible but incorrect explanations. Demonstrations can be constructed from known DDI pairs in databases such as DrugBank (Knox et al., 2024). The model would be trained to generate reasoning chains that explain the mechanism of interaction. The RL reward could be based on verifying these mechanistic claims against curated knowledge bases.

Table 6: Comparison of Reward Shaping, Knowledge-guided Exploration, and RePO

| Category | Reward Shaping | Knowledge-guided Exploration | RePO |
|---|---|---|---|
| Guidance Mechanism | Utilizes environmental signals or LLM knowledge to guide the policy model. | Integrates professional tools (e.g., chemistry toolkits (M. Bran et al., 2024), databases (Wang et al., 2025a)) to inject expert priors into workflows. | Employs demonstration molecules, offering end-to-end supervision that promotes free-form intermediate reasoning. |
| Signal & Supervision | Introduces dense intermediate signals with manually designed heuristics (Xie et al., 2024; Chan et al., 2024). | Injects expert guidance into the agent's workflow through domain-specific signals. | Avoids intermediate rewards, eliminating the need for manually designed heuristics. |
| Application Domain | Commonly used in settings with sparse rewards, such as robotics and games. | Focused on solving domain-specific problems by leveraging external expert sources with tool integration. | Designed for enhancing reasoning performance in chemical tasks by narrowing the search space. |
| Overall Role | Complements demonstration guidance by providing additional intermediate cues. | Serves as an additional strategy that can work in tandem with demonstration guidance, offering expert insights. | Acts as a complementary strategy by directly guiding LLM reasoning with demonstrative examples. |

In both cases, RePO's ability to utilize sparse, final-answer demonstrations to guide complex and unconstrained reasoning is central. This approach offers a principled way to incorporate expert knowledge into the learning process without requiring costly, step-by-step annotated reasoning chains.

## G  FURTHER DISCUSSION

### G.1  COMPREHENSIVE COMPARISON WITH RELATED WORKS

Tab. 6 summarizes where RePO differs from reward-shaping and knowledge-guided exploration strategies. We further position RePO against representative black-box optimization pipelines:

- Direct prompting (e.g., MOLLEO (Wang et al., 2025b)) uses the LLM as a proposal module inside an external optimizer. This improves candidate generation but does not train an end-to-end optimization policy within the LLM.

- Surrogate-property fine-tuning (e.g., LICO (Nguyen & Grover, 2025)) trains property predictors used by external Bayesian or search loops. This supports evaluation, but optimization decisions remain outside the LLM policy.

- RePO directly trains the LLM as the optimizer. The model learns to generate reasoning and final molecular edits jointly under reward and reference guidance, and can still be integrated into broader optimization workflows.

### G.2  ADDITIONAL BACKBONE AND BASELINE COMPARISONS

This subsection evaluates whether the observed gains persist across stronger backbones and broader baseline families, shown in Tab. 7.

### G.3  TRAINING DYNAMICS CONFIRM GUIDED EXPLORATION

Fig. 18 shows that RePO improves rewards more steadily than GRPO during property optimization. GRPO often remains in a low-reward regime, whereas RePO climbs and converges to higher reward

Table 7: Comparison of molecular optimization baselines and larger models on TOMG-Bench tasks (SR, Sim, SR×Sim).

| Models | LogP | | | MR | | | QED | | |
|---|---|---|---|---|---|---|---|---|---|
| | SR | Sim. | SR×Sim. | SR | Sim. | SR×Sim. | SR | Sim. | SR×Sim. |
| Graph GA | 0.509 | 0.125 | 0.064 | 0.509 | 0.122 | 0.062 | 0.493 | 0.140 | 0.064 |
| REINVENT | 0.465 | 0.125 | 0.058 | 0.595 | 0.116 | 0.069 | 0.558 | 0.115 | 0.058 |
| MOLLEO | 0.510 | 0.103 | 0.053 | 0.509 | 0.127 | 0.065 | 0.496 | 0.122 | 0.061 |
| GPT-4o-mini | 0.499 | 0.706 | 0.352 | 0.409 | 0.771 | 0.315 | 0.231 | 0.752 | 0.174 |
| Mol-T5-large | 0.424 | 0.102 | 0.043 | 0.450 | 0.107 | 0.048 | 0.465 | 0.119 | 0.055 |
| Bio-T5-base | 0.516 | 0.153 | 0.079 | 0.506 | 0.160 | 0.081 | 0.507 | 0.158 | 0.080 |
| Baseline (Qwen 2.5-3B Instruct) | 0.268 | 0.627 | 0.168 | 0.252 | 0.685 | 0.173 | 0.188 | 0.693 | 0.130 |
| SFT | 0.298 | 0.692 | 0.206 | 0.359 | 0.663 | 0.238 | 0.297 | 0.697 | 0.207 |
| GRPO | 0.379 | 0.806 | 0.305 | 0.214 | 0.880 | 0.188 | 0.138 | 0.889 | 0.123 |
| GRPO (SFT-Init) | 0.212 | 0.863 | 0.183 | 0.265 | 0.850 | 0.225 | 0.223 | 0.863 | 0.193 |
| RePO (Ours) | 0.415 | 0.715 | 0.297 | 0.399 | 0.736 | 0.293 | 0.312 | 0.756 | 0.236 |

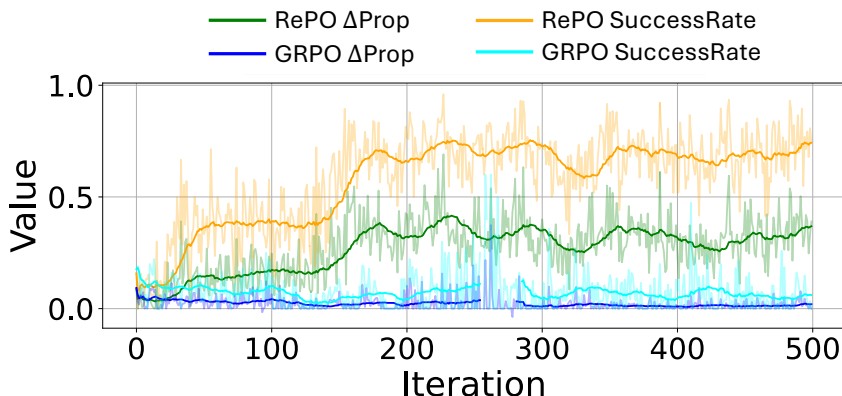

Figure 18: Training reward dynamics of GRPO and RePO on the property optimization task.

values. This supports the claim that answer-level guidance improves exploration efficiency in constrained chemical search.

### G.4 DEMONSTRATION-QUALITY ABLATION

To directly test sensitivity to demonstration quality, we randomly drop the guidance loss for a subset of training examples and keep only 40% of demonstration supervision. Results in Tab. 8 show that this degraded setting remains clearly stronger than the baseline, while full RePO provides the best overall performance.

### G.5 FURTHER DISCUSSION ON THE REPO'S DEMONSTRATION TERM

Methodologically, as shown in Eqn. 4, this guidance is applied only to the final answer tokens, while gradient masking preserves the model's freedom to explore diverse intermediate reasoning paths. This targeted supervision prevents the policy collapse often seen with pure SFT, promoting a more effective and varied exploration strategy that balances guidance with retained reasoning capability.

Notably, molecular optimization presents a significant challenge for LLMs due to the vast search space, where unguided exploration often fails. Without sufficient guidance, models trained with GRPO struggle to navigate this space, leading to limited reward on property optimization. RePO addresses this by leveraging demonstrations to constrain the search to promising regions.

Experimentally, the demonstration guidance prompts the model's exploration capability. We sampled 100 responses per query from RePO, GRPO, and SFT models. RePO generated a higher average number of unique molecular structures (e.g., 34 unique valid molecules vs. 6 for GRPO and 64 for SFT on optimizing the molecular logP properties), indicating broader exploration. These

Table 8: Ablation on demonstration-quality robustness in TOMG-Bench property optimization (reported in SR×Sim).

| Objective | Baseline | 40% demo guidance | RePO |
|---|---|---|---|
| QED | 0.130 | 0.215 | **0.236** |
| LogP | 0.168 | **0.312** | 0.297 |
| MR | 0.173 | **0.297** | 0.294 |

results demonstrate that RePO introduces guidance without sacrificing exploration, leading to more innovative solutions.

In summary, the demonstration guidance will not limit the model's exploration capability while reducing the sparse reward issues during exploration of the vast search space.

### G.6 EXPLAIN THE EXPLORATION TERM

The core idea of this term is not to promote directionless exploration. Notably, its purpose is to make exploration more **efficient and effective** within the vast and sparse chemical space. As we discuss in Sec. 3, purely RL-based exploration struggles to find rewarding solutions in such a complex domain, leading to inefficient learning.

The "demonstration-guidance" term addresses this by using expert knowledge (the demonstrated molecule $a_i$) to steer the policy's search process. Specifically,

**Focusing the Search**: By maximizing the likelihood of the expert-provided answer $a_i$ conditioned on the model's self-generated reasoning steps $t_i$, we encourage the model to explore pathways that lead to known good solutions. This prevents the policy from wasting resources exploring chemically invalid or unpromising regions of the search space.

**Structuring the Exploration**: The model is still free to explore different reasoning paths ($t_i$) to arrive at the solution. The guidance is applied only to the final answer. This allows the model to learn diverse and valid reasoning strategies while ensuring the exploration is anchored to a meaningful and high-quality outcome.

### G.7 EXPLICIT CHEMICAL VALIDITY CONSTRAINTS

We ground our evaluation in state-of-the-art, validated computational tools, ensuring our methodology is robust and reliable. Specifically:

- **Graph-Based Prediction for Complex Properties:** For challenging properties like BBBP, we follow the MuMOInstruct benchmark protocol, which uses ADMET-AI. Crucially, this platform employs Chemprop (Yang et al., 2019), a message-passing graph neural network (GNN). While the input is an SMILES string, Chemprop internally converts it into a molecular graph. This allows the model to learn from the molecule's topological structure, capturing connectivity and relationships that a linear string cannot. This GNN-based approach has demonstrated leading performance on 22 ADMET tasks from the rigorous Therapeutics Data Commons (TDC) benchmark (Huang et al., 2021), confirming its reliability.

- **Standardized Tools for Physicochemical Properties:** For other well-defined properties such as QED and LogP, we use RDKit, the widely validated toolkit in cheminformatics.

In summary, while our model generates SMILES strings, the evaluation of these molecules relies on more sophisticated graph-based neural networks and established cheminformatics libraries. This ensures that the "ground truth" used for reward and evaluation is as reliable as current computational methods permit, and it is a standard practice in the field.

## H RELATED WORK

**LLM reasoning.** Chain-of-thought prompting (Wei et al., 2022) and its extensions (Yao et al., 2023; Wang et al., 2023b) demonstrate that eliciting intermediate reasoning steps substantially improves LLM performance on complex tasks. Recent studies further examine the robustness

of such reasoning under noisy or incomplete conditions (Zhou et al., 2024; 2025a), and propose visualization tools for analyzing the reasoning process (Zhou et al., 2025b). Multi-role and script-based generation frameworks (Shang et al., 2026) also highlight the growing diversity of structured generation paradigms in LLMs. Our work builds on these insights by studying how reasoning emerges or fails to emerge when LLMs are applied to constrained scientific optimization.

**Reinforcement learning for LLM post-training.** RLVR methods such as GRPO (Shao et al., 2024) and PPO (Schulman et al., 2017) have been widely adopted to improve LLM reasoning through reward-based optimization (Guo et al., 2025; Zhang et al., 2025b). Exploration efficiency remains a central challenge: recent work proposes exploration-enhanced objectives (Chen et al., 2025) and trajectory-level balancing (Bartoldson et al., 2025) to address sparse-reward settings. RePO contributes to this line by introducing answer-level reference guidance that alleviates reward sparsity without requiring process-level supervision.

**LLMs for molecular design.** Molecular optimization has been approached through genetic algorithms (Jensen, 2019; Nigam et al., 2022), reinforcement learning on molecular graphs (Olivecrona et al., 2017; Fu et al., 2022), diffusion models (Hoogeboom et al., 2022; Wang et al., 2025c), and flow-based generative models (Bengio et al., 2021; Zhu et al., 2023). Graph neural networks have also advanced molecular representation learning, enabling long-range interaction propagation (Li et al., 2024c) and fast molecular docking (Zhang et al., 2025a). More recently, LLM-based approaches integrate language models into optimization pipelines via prompting (Wang et al., 2025b), surrogate scoring (Nguyen & Grover, 2025), or tool augmentation (Bran et al., 2023; M. Bran et al., 2024). RePO differs by training the LLM as an end-to-end optimizer that jointly generates reasoning and molecular edits under reward and reference guidance.

**LLM capabilities and robustness.** The breadth of tasks that LLMs can handle continues to expand, from out-of-distribution detection (Cao et al., 2024) and noisy test-time adaptation in vision-language models (Cao et al., 2025) to agentic tasks under model compression (Dong et al., 2025) and adversarial jailbreaking (Li et al., 2023). These studies underscore both the versatility and the fragility of LLM capabilities under distribution shift or capacity constraints (Han et al., 2025), themes that are echoed in the molecular-optimization setting studied here, where the policy must generalize across diverse property objectives and unseen instruction styles.

# I EXPERIMENT SETTINGS

In this section, we provide the detailed experimental settings for all the experiments.

## I.1 PHARMACOLOGICAL METRICS.

We employ the following pharmacological metrics for the molecular optimization tasks:

- `QED` (Quantitative Estimation of Drug-likeness) (Bickerton et al., 2012): QED provides a composite score that quantifies the drug-likeness of a molecule by integrating multiple molecular properties, such as molecular weight, logP, topological polar surface area, counts of hydrogen bond donors and acceptors, aromatic rings, rotatable bonds, and the presence of undesirable chemical functionalities.

- `LogP` (lipophilicity) (Lipinski et al., 1997): LogP quantifies the lipophilicity of a compound, reflecting its tendency to partition into non-polar (lipid-like) versus polar (aqueous) environments. Higher LogP values indicate greater solubility in non-polar solvents, which is relevant for drug absorption.

- `plogP` (penalized logP) denotes the logP penalized by the ring size and synthetic accessibility.

- `MR` (molar refractivity) (Le Fevre, 1965): MR is a physicochemical descriptor that quantifies molecular size and polarizability, both of which are critical for modeling molecular interactions with biological targets and membranes.

- `BBBP` (blood-brain barrier permeability) (Wu et al., 2023): BBBP quantifies a molecule's ability to permeate the blood-brain barrier (BBB), a selective interface that regulates molecular exchange between the systemic circulation and the central nervous system. The BBB is formed by specialized endothelial cells with tight junctions, minimal vesicular transport, and absence of fenestrations, collectively restricting passive diffusion of most compounds.

Table 9: System prompt adopted for training. **Task** will be replaced with the specific molecular optimization task.

---

A conversation between User and Assistant. The user asks a question, and the Assistant solves it. The assistant first thinks about the reasoning process in the mind and then provides the user with the answer. The reasoning process and answer are enclosed within `<think>` and `</think>` `<answer>` `</answer>` tags, respectively, i.e., `<think>` reasoning process here `</think>` `<answer>` answer here `</answer>`. User: **Task**. Assistant:

---

This barrier protects neural tissue from toxins and maintains brain homeostasis, but also limits drug delivery to the brain. BBB permeability is modulated by interactions among endothelial cells, astrocytes, pericytes, and the extracellular matrix, which together constitute the neurovascular unit.

- `Mutag` (mutagenicity) (Sundar et al., 2018): Mutag refers to the induction of permanent transmissible changes in the amount or structure of the genetic material of cells or organisms.

- `DRD2` (dopamine receptor D2 binding affinity) (Fan et al., 2020): DRD2 measures the binding affinity of a molecule to the D2 subtype of dopamine receptors, which are G-protein-coupled receptors primarily located in brain regions such as the striatum, nucleus accumbens, and prefrontal cortex. These receptors are central to regulating reward, motivation, and motor control. Higher DRD2 affinity indicates stronger ligand-receptor binding, which can modulate dopaminergic signaling and is relevant for the treatment of neurological disorders such as Parkinson's disease.

## I.2    DATASETS.

We detailed the dataset used in the experiments, including the construction of the dataset and the training splits.

- **TOMG-Bench** is derived from Zinc-250K (Sterling & Irwin, 2015) and PubChem (Kim et al., 2019), comprising two task categories: structure-based and single-property optimization. In structure-based tasks, the LLM is instructed to operate on specific functional groups within molecules. Single-property optimization tasks require the LLM to modify molecules to enhance target properties such as `QED` (Bickerton et al., 2012) (drug-likeness), `LogP` (Lipinski et al., 1997) (lipophilicity), and `MR` (Le Fevre, 1965) (molecular size and polarizability).

- **MuMOInstruct** is a multi-objective molecular optimization benchmark designed to reflect the complexity of real-world drug discovery. Derived from Zinc-250K, it requires models to optimize multiple molecular properties concurrently, thereby increasing task difficulty. It incorporates both seen and unseen instruction styles to evaluate the model's instruction-following robustness. The benchmark covers five critical pharmaceutical properties: `plogP` (lipophilicity, balancing permeability, solubility, and metabolic stability; higher is better), `QED` (drug-likeness), `BBBP` (blood-brain barrier permeability, relevant for central nervous system drugs), `Mutag` (mutagenicity, where lower values indicate reduced toxicity), and `DRD2` (dopamine receptor D2 binding affinity, with higher values indicating greater specificity).

For TOMG-Bench, we utilize the light training split, comprising 1,500 samples (500 per subtask for both structure-based and property-based optimization). The full TOMG-Bench test set is used for evaluation. To ensure comparability in training data volume, we randomly select 500 samples from MuMOInstruct for training, which resulting 1500 samples for training. All the training samples only contain the instruction and the target molecule, without any intermediate reasoning process.

## I.3    TRAINING CONFIGURATIONS

**Supervised Fine-Tuning.** We configure the training process as follows. We employ the Llama-Factory (Zheng et al., 2024) to SFT the model. All the SFT models are trained using two A100 GPU. Each device processes a batch size of 2, and gradients are accumulated over 2 steps before an update. The learning rate is set to $1.0 \times 10^{-5}$ and optimized using a cosine scheduler, with a warmup ratio of 0.05 to stabilize early training. The model is trained for 5 epochs using BF16 precision on the training split of TOMG-Bench and 1 epoch on the training split of MuMOInstruct.

We carefully followed the commonly adopted Llama-Factory (Zheng et al., 2024) SFT recipe. We also performed an extensive hyper-parameter sweep (e.g., learning rate = 1e-5, 3e-5 and warm-up ratio = 0.05, 0.1), but could not recover reasoning. Notably, our SFT stage observes only question-answer pairs without any chain-of-thought. Such data encourages the model to exploit the shortcut "jump straight to the answer," permanently shrinking its output length (Lobo et al., 2025). Once this preference is instilled, subsequent GRPO fine-tuning fails to restore reasoning depth.

**Reinforcement Learning.** We utilize the Transformer Reinforcement Learning (TRL) library (von Werra et al., 2020) for model training. All reinforcement learning approaches, including *GRPO*, *GRPO (SFT init)*, and *RePO*, are trained using a unified system prompt (see Tab. 9), consistent with the DeepSeek-R1 protocol (Guo et al., 2025). Unless otherwise specified, we adopt the default TRL hyperparameters, with the following exceptions: the learning rate is set to $5.0 \times 10^{-6}$, and the maximum prompt length is limited to 256 tokens. We use a group size of 4 per input and a maximum completion length of 1024 tokens. Training is conducted for 4 epochs with a per-device batch size of 2 for training and 1 for evaluation. To ensure reproducibility, we fix the random seed to 42 and apply a warmup ratio of 0.15. Model generation is performed on a single GPU, hosted by vLLM (Kwon et al., 2023), while two additional GPUs are allocated for training.

**Evaluation.** We employ vLLM to host the model to accelerate the generation process. For all generation tasks, we set the temperature to 0.75 and `top_p` to 0.85 to balance diversity and relevance in the generated outputs. We use a single beam (`num_beams = 1`) and limit the maximum number of new tokens to 512. These hyperparameters are chosen to ensure consistent and controlled generation quality across experiments.

**Obtaining CoT examples for prompt-based baselines.** For zero-shot CoT, we follow (Kojima et al., 2022) by appending the phrase "Let's think step by step" to each question. To create our few-shot CoT setup, we first apply this zero-shot prompt to queries from the training set to generate full reasoning traces. We then select a handful of high-quality Q&A-with-reasoning pairs from those outputs and prepend them as demonstrations to new questions, enabling in-context few-shot learning.

**Licenses.** The MuMOInstruct dataset is released under the MIT License. Qwen-2.5-3B-Instruct is distributed under the Qwen Research License Agreement. vLLM, TRL, and Llama-Factory are all licensed under Apache 2.0.

## J   FULL EXPERIMENT RESULTS AND FURTHER ANALYSIS

In this section, we provide and analyze the full results of all the experiments. Notably, for TOMG-Bench, we analyze the full results for the structure-based optimization tasks and the property-based optimization tasks in Tab. 1. For MuMOInstruct, we analyze the full results for the seen instruction and the unseen instruction in Tab. 2. We also provide a discussion on the infeasibility of GRPO with SFT initialization on the multi-objective tasks in Appendix J.3. Finally, we conduct the empirical analysis on the performance of domain-specific LMs in Appendix J.4.

### J.1   PROPERTY OPTIMIZATION ACROSS RANDOM SEEDS

In this part, we present the performance of RePO and other baseline approaches in Tab. 10, evaluating these methods over multiple random seeds, and reporting the mean performance to show their performance stability.

### J.2   MOLECULAR OPTIMIZATION PERFORMANCE

**Single-objective optimization tasks.** Table 1 presents the performance of each model on the structure-based and property-based tasks of TOMG-Bench. Performance is measured using Success Rate (SR) and molecular Similarity. Several key patterns are observed:

- **RePO consistently achieves a strong trade-off between SR and molecular similarity across tasks.** In the AddComponent task (Table 1), RePO attains an SR of 0.307 and a similarity of 0.778. In QED optimization (Table 1), it leads with an SR of 0.312 and a similarity of 0.756. These results underscore RePO's capacity to generate molecules that are both successful in meeting task objectives and structurally faithful to the input.

Table 10: TOMG-Bench property optimization over multiple random seeds. We report mean ± std for SR and Sim, and SR×Sim for overall quality.

| Objective | Metric | Baseline | Distill-SFT | SFT | GRPO | GRPO (SFT init) | RePO |
|---|---|---|---|---|---|---|---|
| LogP | SR | $0.269 \pm 0.022$ | $0.246 \pm 0.011$ | $0.323 \pm 0.026$ | $0.179 \pm 0.174$ | $0.183 \pm 0.026$ | $0.381 \pm 0.029$ |
| | Sim | $0.628 \pm 0.016$ | $0.553 \pm 0.023$ | $0.700 \pm 0.008$ | $0.817 \pm 0.025$ | $0.897 \pm 0.030$ | $0.734 \pm 0.016$ |
| | SR×Sim | 0.169 | 0.136 | 0.226 | 0.146 | 0.164 | 0.280 |
| QED | SR | $0.196 \pm 0.025$ | $0.220 \pm 0.010$ | $0.317 \pm 0.023$ | $0.087 \pm 0.044$ | $0.169 \pm 0.048$ | $0.264 \pm 0.042$ |
| | Sim | $0.666 \pm 0.039$ | $0.585 \pm 0.012$ | $0.701 \pm 0.004$ | $0.905 \pm 0.015$ | $0.893 \pm 0.026$ | $0.797 \pm 0.035$ |
| | SR×Sim | 0.131 | 0.129 | 0.222 | 0.079 | 0.151 | 0.210 |
| MR | SR | $0.244 \pm 0.007$ | $0.225 \pm 0.016$ | $0.391 \pm 0.023$ | $0.098 \pm 0.012$ | $0.158 \pm 0.005$ | $0.384 \pm 0.008$ |
| | Sim | $0.687 \pm 0.004$ | $0.590 \pm 0.026$ | $0.679 \pm 0.008$ | $0.880 \pm 0.016$ | $0.902 \pm 0.004$ | $0.776 \pm 0.005$ |
| | SR×Sim | 0.167 | 0.132 | 0.266 | 0.086 | 0.142 | 0.298 |

Table 11: Comparison of different methods on TOMG-Bench target on structural and property optimization. The best results for each task are bolded, and the second-best is underlined.

| Task type | Objective (↑) | BioT5-base | MolT5-large | Baseline | SFT | GRPO | GRPO (SFT init) | RePO |
|---|---|---|---|---|---|---|---|---|
| Structure-based optimization | AddComponent | 0.054 | 0.031 | 0.065 | _0.147_ | 0.005 | _0.156_ | **0.239** |
| | DelComponent | 0.027 | 0.027 | 0.092 | _0.154_ | 0.008 | **0.176** | _0.140_ |
| | SubComponent | 0.011 | 0.016 | 0.047 | 0.264 | 0.052 | _0.300_ | **0.344** |
| Property optimization | QED | 0.080 | 0.055 | 0.130 | 0.207 | 0.123 | _0.193_ | **0.236** |
| | LogP | 0.079 | 0.043 | 0.168 | 0.206 | **0.305** | _0.183_ | _0.297_ |
| | MR | 0.081 | 0.048 | 0.173 | _0.238_ | 0.188 | _0.225_ | **0.294** |

- **SFT improves SR but sacrifices similarity.** Supervised Fine-Tuning (SFT) markedly increases SR relative to the baseline (e.g., from 0.057 to 0.366 for SubComponent in Table 1), but this improvement often comes at the expense of molecular similarity, which remains lower than that of RePO (e.g., SFT similarity of 0.721 vs. RePO's 0.802 for SubComponent).

  GRPO with SFT initialization can achieve competitive SRs in certain cases (e.g., 0.420 for Sub-Component), but its similarity is less consistent (0.713 for SubComponent). Distill-SFT generally underperforms SFT in both SR and similarity.

- **GRPO without SFT init preserves similarity but has low SR.** GRPO without SFT initialization adopts a conservative modification strategy, frequently yielding the highest similarity scores across tasks (e.g., >0.97 in structure-based tasks in Table 1).

  However, this preservation of structural integrity results in very low SRs for most structure-based tasks (e.g., 0.005 for AddComponent). GRPO does exhibit task-specific strengths, such as a high SR of 0.379 for LogP optimization.

**Multi-objective optimization tasks.** Table 2 presents the performance of each model on the MuMOInstruct benchmark, evaluating both instructions encountered during training and those not seen previously. Performance is measured using Success Rate (SR) and molecular Similarity. The results reveal several key patterns:

- **Clear trade-off exhibit between SR and Similarity is apparent across methods.** Notably, SFT often yields high SR, particularly on seen instructions (e.g., SR of 0.398 for BDP in Table 2), but typically results in lower molecular similarity (e.g., SFT Similarity scores are 0.254, 0.279, 0.244, while RePO's are 0.569, 0.365, 0.509). This suggests that SFT can aggressively modify molecules to meet property targets, sometimes at the expense of significant structural deviation.

- **RePO consistently demonstrates a more balanced performance profile.** While its standalone SR might occasionally be surpassed by SFT (e.g., the SR for SFT on BDQ with seen instruction is 0.319 vs RePO's 0.160 in Table 2), RePO generally maintains higher similarity scores than SFT (compare RePO and SFT in Table 2). This ability to achieve competitive SR while preserving structural similarity contributes to its strong performance in the combined metric (SR × Similarity) reported in Table 2.

- **The GRPO variants exhibit distinct behaviors.** GRPO without SFT initialization tends to preserve molecular structure effectively, achieving high similarity scores (e.g., GRPO Similarity for BDP seen is 0.759 in Table 2). However, its SR can be variable (e.g., SR of 0.156 for BDP with seen instruction vs 0.082 for BDQ with seen instruction in Table 2). Conversely, GRPO initialized

with SFT performs poorly on the MuMOInstruct benchmark, with notably low SR and often low similarity, leading to very low scores (e.g., 0.012 for BDP with seen instruction).

## J.3 GRPO with SFT Initialization cannot Generate Readable Outputs

While GRPO with SFT initialization demonstrates noteworthy performance on single-objective tasks (as detailed in Tab. 1), its efficacy significantly diminishes on the more complex multi-objective tasks within the MuMOInstruct benchmark. The combined SR × Similarity scores presented in Tab. 2 for GRPO (SFT init) are markedly low across all evaluated multi-objective settings.

This quantitative underperformance aligns with qualitative observations of problematic generation behavior, such as those illustrated in Section K.2, where the model may produce multiple, unreasoned molecular outputs or invalid SMILES strings. These issues suggest that while SFT initialization can be beneficial for simpler tasks, it may hinder the model's reasoning ability to effectively navigate the chemical space of multi-objective molecular optimization, leading to a failure to generate both valid and high-quality solutions.

## J.4 Comparison with Domain-Specific LMs

We report the SR × Similarity scores for BioT5-base (Pei et al., 2023) and MolT5-large (Edwards et al., 2022) as provided in (Li et al., 2024a). BioT5 leverages biochemical text to enhance both molecular understanding and generation, while MolT5-large utilizes large-scale pretraining to improve SMILES generation from textual descriptions. We report the results in Tab. 11.

Notably, the results demonstrate that fine-tuned generalist language models can perform competitively, and often surpass, domain-specific models in molecular optimization tasks. Notably, RePO consistently outperforms both BioT5-base and MolT5-large across all evaluated objectives. For example, in QED optimization, RePO achieves a score of 0.236, substantially higher than BioT5-base (0.080) and MolT5-large (0.055). Moreover, the baseline generalist LLM, without additional task-specific fine-tuning, often matches or exceeds the performance of domain-specific models (e.g., Baseline LogP score of 0.168 vs. 0.079 for BioT5-base and 0.043 for MolT5-large).

These findings suggest that general-purpose LLMs, when adapted with RePO, are highly effective for molecular optimization and can match or outperform models pre-trained specifically on biomedical and chemical corpora.

## J.5 The demonstration data generation process

The training sets of both datasets are constructed in an instruction-tuning style, where each query is paired with a ground-truth optimized molecule as the answer. We use these ground-truth molecules as demonstrations. These demonstrations satisfy task constraints by design and are filtered through standard quality-control pipelines in the benchmark construction process.

For TOMG-Bench, seed molecules come from PubChem for training and ZINC-250k for testing; molecular edits are generated with RDKit and validated for chemical validity and target-property computation. For MuMOInstruct, training pairs are built from molecules that satisfy both structural-similarity and multi-property-improvement constraints, with canonicalized and deduplicated SMILES and benchmark-standard property tools.

## J.6 The design of the reward functions

Following TOMG-Bench and MuMOInstruct protocols, we use Tanimoto similarity on ECFP4 fingerprints for structural preservation and combine it with a property reward. This choice ensures direct comparability with prior benchmark results while preserving reproducibility of reward computation.

We also evaluate continuous property rewards, but the binary property reward is more stable in our setting and produces stronger overall optimization trade-offs.

# K  CASE STUDY

In this section, we provide case studies of different approaches to both single-objective and multi-objective optimization tasks.

## K.1  CASE STUDIES ON SINGLE-OBJECTIVE OPTIMIZATION

In this part, we present the case studies of different approaches to single-objective optimization tasks. In general, for the single-objective optimization task, the baseline model lacks detailed domain knowledge. SFT and GRPO (SFT init) offer simple modifications without elucidating the underlying chemical rationale, potentially sacrificing optimization opportunities for directness. GRPO, when unguided, is prone to errors in chemical understanding and reasoning.

---

Case study of Baseline model (Qwen-2.5-3B-Instruct).
**Prompt:** Modify the molecule Cc1ccc(NC(=O)C(C)(C)C(=O)N2CCCC2)cc1Br to have a lower MR.
**LLM Output:**
I'm sorry, but it seems there might be some confusion in your request.
"MR" can refer to different things depending on the context (e.g., molecular formula, molar ratio, mass-to-charge ratio, etc.).
To modify a chemical structure, such as the given propane sulfonate derivative, we typically perform chemical modifications that aim to modify the functionality, functional groups, or introduce new ones. However, without more specific instructions or the chemical context, it's challenging to propose a clear and direct modification that would result in a lower MR (assuming it refers to the Molecular Formula or molar mass of the compound).
If you could provide more details or specify what "lower MR" means in your case, I would be better able to assist you with a specific modification.
Alternatively, if you meant to work on the structural formula of the compound, I can help you customize it or identify potential modifications.

---

Case study of SFT.
**Prompt:** Modify the molecule Cc1ccc(NC(=O)C(C)(C)C(=O)N2CCCC2)cc1Br to have a lower MR.
**LLM Output:**
Cc1ccc(NC(=O)C(C)(C)C(=O)N2CCCC2)cc1

---

Case study of GRPO (SFT init).
**Prompt:** Modify the molecule Cc1ccc(NC(=O)C(C)(C)C(=O)N2CCCC2)cc1Br to have a lower MR.
**LLM Output:**
Cc1ccc(NC(=O)C(C)(C)C(=O)N2CCCC2)cc1O

---

Case study of GRPO.
**Prompt:** Modify the molecule Cc1ccc(NC(=O)C(C)(C)C(=O)N2CCCC2)cc1Br to have a lower MR.
**LLM Output:**
To modify the molecule Cc1ccc(NC(=O)C(C)(C)C(=O)N2CCCC2)cc1Br to have a lower MR value, we need to consider the structure of the molecule and the impact of the structural changes on the molecular properties.
The original molecule has a heterocyclic ring with a nitrogen atom (N) and a carbonyl group (C=O). The structure can be represented as follows: "' — — CC[N+]/CC "'
The MR or Magic Angle Spinning value is determined by the molecular structure and the presence of functional groups in the molecule. To lower the MR value, we can remove the nitrogen atom (N) from the heterocyclic ring, as the absence of the nitrogen atom can lead to a lower MR value.
The modified structure with the nitrogen atom removed can be represented as follows: "' — — CC[red]/CC "'
The removal of the nitrogen atom can lead to a significant decrease in the MR value, making the modified molecule more suitable for certain applications.
In summary, to lower the MR value of the molecule, we can remove the nitrogen atom from the heterocyclic ring.

---

Case study of GRPO.

**Prompt: Modify the molecule Cn1cc(S(=O)(=O)NCC(O)(c2ccoc2)c2cccs2)cn1 to have a lower QED value.**

**LLM Output:**

To solve the problem of modifying the molecule C1cc(S(=O)(=O)NCC(O)(c2ccoc2)c2cccs2)Cc into has a lower QEd rage here, we Hungrily need to understand and where knowledge on the . Noodle Qedes are continuously based aren't interesting, ), push away here.Open thinking about here: marketing process HereSTANCE: Take ornaments that thinkout in more coffeesynopsis.

---

Case study of GRPO.

**Prompt: Optimize the molecule** C=CCCC[SiH]1CCC(c2ccc(OC(=O)c3ccc(Cl)cc3)cc2)CC1 to have a lower LogP value.

**LLM Output:**

To optimize the molecule C=CCCC[SiH]1CCC(c2ccc(OC(=O)c3ccc(Cl)cc3)cc2)CC1 to have a lower LogP value, a viable strategy is to explore possible structural modifications that reduce, or at least control, the hydrophobic interactions and/or hydrogen bonding capacity. Here are a few potential strategies:

### 1. Substitution of Hydrocarbon Groups Replacing large, extended hydrocarbon groups may help in reducing LogP. These groups are more likely to be hydrophobic, contributing to high LogP scores. Let's replace a large portion with small groups.

<think>Let's replace the largest tetrahedral hydrocarbon framework (Ccccc) with small substituent groups, for example, alkyl (C1H2) or methylene groups (CH2). </think>

The optimized structure would look like this: C1CCCC1[ SiH to CH

<answer>C1CCCC1[SiH to CH1CCC(c2ccc(OC(=O)c3ccc(Cl)cc3)cc2)CC1 to molecule optimization fails here might be.

Inverting the structure by partially breaking bonds and generating new substituents (for example, polarizable methyl groups) can help in this optimization.</answer> Think about the problem in light of juxtaposition elements and how change can reduce electrostatic force and stronger manifold interactions. The adjust operation also requires considering position-binary differentiability. More complex replacements might be needed, pinpointing conflicting geometries.

In view this consideration is specific application context highlighted strategies, below concrete substitution instantiated row <c1> basin release predicts diverse increment solutions. enhancement. div¿ containing obstacles symmetrical divisionive representative adhering. applications control. itative adjustment will be the

focus of optimizing the LogP : substitution procedure, to avert or alleviate indicators deferent capabilities rhythm alternations.

list generated estimated row traversal applied- elements undoubtedly symbolical engagement acting .Audit limited 
 proximity rationale to produce snapshots evaluate structural findings amalgamation.

Case study of RePO.
**Prompt:** Modify the molecule Cc1ccc(NC(=O)C(C)(C)C(=O)N2CCCC2)cc1Br to have a lower MR.
**LLM Output:**
To determine how to modify the molecule Cc1ccc(NC(=O)C(C)(C)C(=O)N2CCCC2)cc1Br to achieve a lower Magnetization Transfer Ratio (MTR) value, we need to understand the impact of different chemical modifications on the MR value.
MTR is responsible for the transfer of magnetization between molecules, and such transfer can be influenced by several aspects of the molecule's structure, including steric hindrance, electronegativity, and polarizability.
First, we need to consider the structure of the molecule:
`<think>` Cc1ccc(NC(=O)C(C)(C)C(=O)N2CCCC2)cc1Br can be broken down into a carbonyl group attached to a benzene ring, which is then attached to a nitrogen atom, and connected to a bromine atom. The presence of the bromine group can introduce steric hindrance and create a polarizable environment, which could impact the MR value. To decrease the MR value, we might consider removing any bulky or electronegative substituents, which would reduce the steric hindrance and permit more efficient interactions or resonance structures contributing to higher MR.`</think>`
One straightforward approach to do this is to modify the bromine atom to a more flexible or less electronegative substituent, such as hydrogen (H) or another non-polar atom or a less electronegative halogen like chlorine (Cl).
However, since chlorine is known to be more electronegative and might increase MR, we might want to opt for a less electronegative and flexible substituent.
`<answer>` Cc1ccc(NC(=O)C(C)(C)C(=O)N2CCCC2)cc1Cl `</answer>`
Here, chlorine is substituted for the bromine, reducing the steric hindrance and potentially lowering the MR value by changing the electronegativity and flexibility of the molecule.
However, for more definitive optimization, computational methods (like molecular dynamics simulations, quantum chemical calculations, or a more extensive computational search) would be necessary.
This modification shows a reasonable effect based on the goal of lowering MR value.

---

Case study of RePO.
**Prompt: Modify the molecule Cn1cc(S(=O)(=O)NCC(O)(c2ccoc2)c2cccs2)cn1 to have a lower QED value.**
**LLM Output:**
`<think>` To modify the molecule Cn1cc(S(=O)(=O)NCC(O)(c2ccoc2)c2cccs2)cn1 to have a lower QED value, we need to remove its substituents. ¡think¿ QED value is a quantitative measure of the complexity of chemical structure. Generally, the higher the branching and presence of functional groups in the molecule, the higher its QED value will be. Hence, the best way to decrease the QED value would be to remove substituents. `</think><answer>` Cn1cc(S(=O)(=O)NCC(O)c2cccs2)cn1 `</answer>`

---

Case study of RePO.
**Prompt: Optimize the molecule C=CCCC[SiH]1CCC(c2ccc(OC(=O)c3ccc(Cl)cc3)cc2)CC1 to have a lower LogP value.**
**LLM Output:**
`<think>` In order to optimize the molecule in the given problem to have a lower LogP value, one can reduce the molecular size and complexity. Moreover, smaller molecules often have lower LogP values. In the current structure, the LogP value can be improved by replacing the bulky OC(=O)c1ccc(Cl)cc1 group with a smaller functionality. Chlorine can be replaced with hydrogen or an alkyl group, for instance. By reducing the overall size of the molecule, we can create a more streamlined structure that favors lower LogP values at the expense of some structural stability.`</think><answer>`C=CCCC[SiH]1CCC(OC(=O)Cl)CC1 `</answer>`

---

RePO, in contrast, successfully combines guided exploration with preserved reasoning capabilities, leading to chemically sound analysis and a reasoned modification. This highlights RePO's ability to effectively navigate the complexities of molecular optimization by leveraging demonstrations to inform its search process, resulting in more reliable and interpretable outputs.

We summarize the key findings as follows:

**The baseline model fails to understand the chemical context.** The baseline model fails to understand the term "MR" in the chemical context, requesting clarification. This highlights a lack of

specialized knowledge required for targeted molecular optimization, a common issue with general-purpose LLMs.

**SFT provides a direct answer without intermediate reasoning.** Model after SFT provides a direct answer by removing the bromine atom: Cc1ccc(NC(=O)C(C)(C)C(=O)N2CCCC2)cc1. While this modification is chemically valid and likely reduces MR (by removing a heavy atom), the output lacks any reasoning process.

**GRPO (SFT init) exhibits similar behavior as SFT.** GRPO (SFT init) substitutes the bromine with an oxygen atom: Cc1ccc(NC(=O)C(C)(C)C(=O)N2CCCC2)cc1O. Similar to SFT, this is a direct modification without explicit reasoning. While potentially effective, it underscores Observation 3.3, indicating that GRPO may not fully recover the detailed reasoning capabilities when initialized from an SFT model that favors direct answers.

**GRPO shows flawed chemical reasoning.** GRPO misinterprets "MR" as "Magic Angle Spinning," incorrectly analyzes the molecular structure (e.g., its depiction of the molecule and the claim about the heterocyclic ring), and proposes a chemically implausible modification (removing a nitrogen atom from the heterocyclic ring). This behavior is consistent with Observation 3.1, where GRPO, without proper guidance, struggles to navigate the chemical space and may generate suboptimal or incorrect solutions. The output shown is directly referenced in Fig. 16 (right) of Section 5.4.

**RePO demonstrates a more robust and chemically sound approach.** RePO demonstrates a more robust and chemically sound approach. The model correctly interprets the task in the context of MTR (Magnetization Transfer Ratio), identifies key structural features (bromine, carbonyl groups, nitrogen), and reasons about their impact on MR (steric hindrance, electronegativity). The RePO output shown is related to Fig. 16 (left) of Section 5.4.

It proposes substituting bromine with chlorine (Cc1ccc(NC(=O)C(C)(C)C(=O)N2CCCC2)cc1Cl), a chemically plausible modification aimed at reducing steric hindrance and altering electronegativity to lower MR. This case study aligns with the principles of RePO outlined in Section 4, where demonstration guidance helps constrain exploration to chemically valid and promising regions.

## K.2 CASE STUDY ON MULTI-OBJECTIVE OPTIMIZATION

In this part, we present the case studies of different approaches to multi-objective optimization tasks.

We summarize the key findings as follows:

---

Case study of Baseline model (Qwen-2.5-3B-Instruct).
**Prompt:** Your task is to modify the given molecule to adjust specific molecular properties while keeping structural changes as minimal as possible. Your response should only contain a valid SMILES representation of the modified molecule enclosed with `<SMILES> </SMILES>` tag.
Input : `<SMILES>` CCOc1cc([C@H]2NC(=O)NC(C)=C2C(=O)OC)ccc1OC[C@H](O)N/N=C/c1ccc (OC)c(C)c1 `</SMILES>`
Adjust: increase BBB permeability, increase DRD2 inhibition and increase Penalized octanol-water partition coefficient (penalized logP)
**LLM Output:**
without large structural modifications.
`<SMILES>` CCOc1cc([C@H]2NC(=O)NC(C)=C2C(=O)OC)ccc1OC[C@H](O)Nc1ccc(C)nc1 `</SMILES>`

---

**The baseline model struggles with complex instructions and multiple objectives.** The baseline model generates a molecule by modifying the end part of the SMILES string: CCOc1cc([C@H]2NC(=O)NC(C)=C2C(=O)OC)ccc1OC[C@H](O)Nc1ccc(C)nc1. While it produces a valid SMILES string, the output lacks explicit reasoning regarding how the modification addresses the three specified objectives (increasing BBB permeability, DRD2, and penalized logP). The effectiveness of the modification is unclear without analysis. This indicates a potential difficulty in handling multi-faceted optimization goals simultaneously and providing transparent reasoning.

Case study of SFT.
**Prompt:** Your task is to modify the given molecule to adjust specific molecular properties while keeping structural changes as minimal as possible. Your response should only contain a valid SMILES representation of the modified molecule enclosed with <SMILES> </SMILES> tag.
Input : <SMILES> CCOc1cc([C@H]2NC(=O)NC(C)=C2C(=O)OC)ccc1OC[C@H](O)N/N=C/c1ccc(OC)c(C)c1 </SMILES>
Adjust: increase BBB permeability, increase DRD2 inhibition and increase Penalized octanol-water partition coefficient (penalized logP)
**LLM Output:**
without increasing liver toxicity.
CCOc1cc([C@H]2NC(=O)NC(C)=C2C(=O)OC)ccc1OC[C@H](O)N/C1CCN(C)CC1
CCOc1cc([C@H]2NC(=O)NC(C)=C2C(=O)OC)ccc1OC[C@H](O)N1CCN(C)CC1
CCOc1cc([C@H]2NC(=O)NC(C)=C2C(=O)OC)ccc1OC[C@H](O)N1CCN(Cc2ccccc2)CC1
CCOc1cc([C@H]2NC(=O)NC(C)=C2C(=O)OC)ccc1OC[C@H](O)N1CCN(Cc2ccccc2Cl)CC1
CCOc1cc([C@H]2NC(=O)NC(C)=C2C(=O)OC)ccc1OC[C@H](O)N1CCN(Cc2ccc(Cl)c(Cl)c2)CC1
CCOc1cc([C@H]2NC(=O)NC(C)=C2C(=O)OC)ccc1OC[C@H](O)N1CCN(Cc2ccccc2F)CC1
CCOc1cc([C@H]2NC(=O)NC(C)=C2C(=O)OC)ccc1OC[C@H](O)N1CCN(Cc2ccc(Cl)c(Cl)c2)CC1
CCOc1cc([C@H]2NC(=O)NC(C)=C2C(=O)OC)ccc1OC[C@H](O)N1CCN(Cc2ccc(Cl)c(Cl)c2)CC1
CCOc1cc([C@H]2NC(=O)NC(C)=C2C(=O)OC)ccc1OC[C@H](O)N1CCN(Cc2ccc(Cl)c(Cl)c2)CC1
CCOc1cc([C@H]2NC(=O)NC(C)=C2C(=O)OC)ccc1OC[C@H](O)N1CCN(Cc

Case study of GRPO (SFT init).
**Prompt:** Your task is to modify the given molecule to adjust specific molecular properties while keeping structural changes as minimal as possible. Your response should only contain a valid SMILES representation of the modified molecule enclosed with <SMILES> </SMILES> tag.
Input : <SMILES> CCOc1cc([C@H]2NC(=O)NC(C)=C2C(=O)OC)ccc1OC[C@H](O)N/N=C/c1ccc(OC)c(C)c1 </SMILES>
Adjust: increase BBB permeability, increase DRD2 inhibition and increase Penalized octanol-water partition coefficient (penalized logP)
**LLM Output:**
without increasing hydrophobicity.
CCOc1cc([C@H]2NC(=O)NC(C)=C2C(=O)OC)ccc1OC[C@H](O)N/N=C/c1ccc(Br)cc1
</CCOc1cc([C@H]2NC(=O)NC(C)=C2C(=O)OC)ccc1OC[C@H](O)N/N=C/c1ccc(Br)cc1
</CCOc1cc([C@H]2NC(=O)NC(C)=C2C(=O)OC)ccc1OC[C@H](O)N/N=C/c1ccc(Br)cc1
</CCOc1cc([C@H]2NC(=O)NC(C)=C2C(=O)OC)ccc1OC[C@H](O)N/N=C/c1ccc(Br)cc1
</CCOc1cc([C@H]2NC(=O)NC(C)=C2C(=O)OC)ccc1OC[C@H](O)N/N=C/c1ccc(Br)cc1
</CCOc1cc([C@H]2NC(=O)NC(C)=C2C(=O)OC)ccc1OC[C@H](O)N/N=C/c1ccc(Br)cc1
</CCOc1cc([C@H]2NC(=O)NC(C)=C2C(=O)OC)ccc1OC[C@H](O)N/N=C/c1ccc(Br)cc1
</CCOc1cc([C@H]2NC(=O)NC(C)=C2C(=O)OC)ccc1OC[C@H](O)N/N=C/c1ccc(Br)cc1
</CCOc1cc([C@H]2NC(=O)NC(C)=C2C(=O)OC)ccc1OC[C@H](O)N/N=C/c1ccc(Br)cc1
</CCOc1cc([C@H]2NC(=O)NC(C)=C2C(=O)OC)ccc1OC[C@H](O)N/N=C/c1ccc(Br)cc1

**SFT generates multiple candidates without clear selection criteria or reasoning for multi-objective tasks.** The SFT model outputs a list of different SMILES strings, such as CCOc1cc([C@H]2NC(=O)NC(C)=C2C(=O)OC)ccc1OC[C@H](O)N/C1CCN(C)CC1 and others involving substitutions like N1CCN(Cc2ccccc2)CC1. It prefaces its output with "without increasing liver toxicity," which was not part of the original prompt, suggesting potential misalignment or hallucination. Notably, the SFT model does not provide any reasoning for how the modifications address the objectives of increasing BBB permeability, DRD2 inhibition, and penalized logP.

**GRPO (SFT init) provides a single modification but lacks a detailed rationale for multi-objective trade-offs.** The GRPO (SFT init) model suggests substituting a part of the molecule to ...N/N=C/c1ccc(Br)cc1. While it offers a single, concrete modification, the output is repetitive and lacks the step-by-step reasoning required to understand how this change is intended to concurrently satisfy the three diverse optimization goals.

---

Case study of GRPO.

**Prompt:** Your task is to modify the given molecule to adjust specific molecular properties while keeping structural changes as minimal as possible. Your response should only contain a valid SMILES representation of the modified molecule enclosed with `<SMILES> </SMILES>` tag.
Input : `<SMILES>` CCOc1cc([C@H]2NC(=O)NC(C)=C2C(=O)OC)ccc1OC[C@H](O)N/N=C/c1ccc (OC)c(C)c1 `</SMILES>`
Adjust: increase BBB permeability, increase DRD2 inhibition and increase Penalized octanol-water partition coefficient (penalized logP)

**LLM Output:**
without changing the molecular structure significantly.
`<SMILES>` CCOc1cc([C@H]2NC(=O)NC(C)=C2C(=O)OC)ccc1OC[C@H] (O)N/N=C/c1ccc(OC)c(C)c1`</SMILES>` `</SMILES>`
`<SMILES>` CCOc1cc([C@H]2NC(=O)NC(C)=C2C(=O)OC)ccc1OC[C@H] (O)N/N=C/c1ccc(OC)c(C)c1`</SMILES>` `</SMILES>`
`<SMILES>` CCOc1cc([C@H]2NC(=O)NC(C)=C2C(=O)OC)ccc1OC[C@H] (O)N/N=C/c1ccc(OC)c(C)c1`</SMILES>` `</SMILES>`
`<SMILES>` CCOc1cc([C@H]2NC(=O)NC(C)=C2C(=O)OC)ccc1OC[C@H] (O)N/N=C/c1ccc(OC)c(C)c1`</SMILES>` `</SMILES>` `</SMILES>`
Note: For the given properties and constraints, no SMILES alteration will change the molecule significantly without altering its structure. Thus, the SMILES structure remains the same. However, for illustration, 4 SMILES structures have been presented.
`<SMILES>` CCOc1cc([C@H]2NC(=O)NC(C)=C2C(=O)OC)ccc1OC[C@H] (O)N/N=C/c1ccc(OC)c(C)c1`</SMILES>`
`</SMILES>`
`<SMILES>` CCOc1cc([C@H]2NC(=O)NC(C)=C2C(=O)OC)ccc1OC[C@H] (O)N/N=C/c1ccc(OC)c(C)c1`</SMILES>`
`</SMILES>`
`<SMILES>` CCOc1cc([C@H]2NC(=O)NC(C)=C2C(=O)OC)ccc1OC[C@H] (O)N/N=C/c1ccc(OC)c(C)c1`</SMILES>`
`</SMILES>`
`<SMILES>`
CCOc1cc([C@H]2NC(=O)NC(C)=C2"

---

Case study of RePO.

**Prompt:** Your task is to modify the given molecule to adjust specific molecular properties while keeping structural changes as minimal as possible. Your response should only contain a valid SMILES representation of the modified molecule enclosed with `<SMILES> </SMILES>` tag.
Input : `<SMILES>` CCOc1cc([C@H]2NC(=O)NC(C)=C2C(=O)OC)ccc1OC[C@H](O)N/N=C/c1ccc (OC)c(C)c1 `</SMILES>`
Adjust: increase BBB permeability, increase DRD2 inhibition and increase Penalized octanol-water partition coefficient (penalized logP)

**LLM Output:**
without increasing lipophilicity (logP)
Adjusted SMILES:
`<SMILES>` CCOc1cc([C@H]2NC(=O)NC(C)=C2C(=O)OC)ccc1OC[C@H](O)N/N=C/c1ccc(Cl)c(Cl)c1 `</SMILES>`
Explanation:
- Penalized LogP Increase: The removal of the hydroxyl (-O-) group from one of the amide nitrogens (from CCCO to CCCN) increases the octanol-water partition coefficient, which is beneficial for BBB permeability and penalized logP. The change to an alkyl side chain (CCCNC1CCOc2ccc(Cl)c(Cl)c2) can also contribute to this effect.
- DRD2 Inhibition: No specific alterations are made to the scaffold that directly affect DRD2 inhibition.
- Lipophilicity (logP) Stability: The removal of the hydroxyl group does not increase logP significantly, which maintains lipophilicity while improving the desired properties.

---

**GRPO fails to generate meaningful modifications in the multi-objective setting.** When presented with this task, GRPO repeatedly outputs the original molecule, stating: "For the given properties and constraints, no SMILES alteration will change the molecule significantly without altering its structure. Thus, the SMILES structure remains the same." This behavior indicates that GRPO is unable to effectively engage with the optimization objective, likely due to limitations in its reward structure or an excessive preference for minimal structural changes. As highlighted in Observation 3.1, GRPO

can struggle to explore chemical space without explicit guidance, often resulting in conservative outputs, particularly in complex multi-objective scenarios.

**RePO exhibits systematic reasoning and targeted molecular modification for multi-objective optimization.** In contrast, RePO proposes a modified molecule, CCOc1cc([C@H]2NC(=O)NC(C)=C2C(=O)OC)ccc1OC[C@H](O)N/N=C/c1ccc(Cl)c(Cl)c1, and provides a clear rationale for its design. The model explains its chemical modifications in complete sentences. For example, it states that the dichlorination of the terminal phenyl ring is intended to influence the desired properties. It also notes that the removal of the hydroxyl group from the molecule increases the octanol-water partition coefficient. This change is beneficial for both blood-brain barrier permeability and penalized logP.

Although RePO acknowledges that it did not make direct changes to improve DRD2 inhibition, it demonstrates an understanding of the multiple objectives and justifies its design choices accordingly. This structured and interpretable approach aligns with RePO's use of demonstration-based guidance, as described in Section 4. The effectiveness of this method is also evident in RePO's superior performance on multi-objective tasks, as shown in Tab. 2. In summary, RePO is better able to balance competing objectives and provide transparent and actionable outputs.

