# OpenReview forum: "Reference-guided Policy Optimization for Molecular Optimization via LLM Reasoning"
_ICLR.cc/2026/Conference — ICLR 2026 Poster_

### Official Review · Reviewer_Aw6o · 2025-10-28

**Soundness:** 3
**Presentation:** 3
**Contribution:** 3
**Rating:** 4
**Confidence:** 3

**Summary:**

The paper presents a new method called DePO which was modified from standard RLVR procedure via using a demonstration-guided term in the objective function. It provides a novel framework to LLM-driven molecular optimization and through experimental results, demonstrates that the method can enhance reasoning and task performance.

**Strengths:**

The paper is well written and the experimental design is robust. It also provides a detailed analysis of current drawbacks on RLVR and how to address them via modification on the architecture, algorithm and reward function. The results demonstrate good improvement compared to the baseline and support the paper's aim of providing better reasoning in molecular optimisation.

**Weaknesses:**

The paper uses only Qwen-2.5-3B Instruct as a single backbone for the baseline and base model (and to demonstrate scaling also use Qwen-2.5-7B Instruct). The results and findings could be more robust if it has had demonstrated results using other models since the base reasoning and optimisation capacity can be different. This would be useful to determine whether DePO still offer improvement to the other methods (while using the same base model).

**Questions:**

Is there any disadvantage with constraining the exploration space in DePO?
Table 1 and 2 show the overall results on on TOMG-Bench. If that is an average performance over the dataset, what is the range of errors in addition to the mean and do they vary  or stay similar across the methods?

---

> ### Author Response · Authors · 2025-11-27
> **Response to Reviewer Aw6o (part 1/3)**
>
> **We thank Reviewer Aw6o for the constructive feedback.** In this rebuttal, we address your concerns about (i) whether DePO’s gains persist beyond the Qwen family and (ii) the potential disadvantages of constrained exploration and the variability of our reported results.
> - First, we add experiments on an architecturally different backbone (Llama-3.1-8B-Instruct) under the same TOMG-Bench setup, showing that DePO continues to outperform the backbone and GRPO/SFT baselines.
> - Second, we clarify the tradeoffs of similarity-constrained exploration, explain when it may be limiting, and provide multi-seed results (mean $\pm$ standard deviation) demonstrating that DePO’s improvements are stable and larger than the corresponding error bars.
>
>
> Point-by-point responses are provided below, together with the revisions we will incorporate into the manuscript.
>
> ### W1: Experiment on the model family beyond Qwen.
> > The paper uses only Qwen-2.5-3B Instruct as a single backbone for the baseline and base model (and to demonstrate scaling also use Qwen-2.5-7B Instruct). The results and findings could be more robust if it has had demonstrated results using other models since the base reasoning and optimisation capacity can be different. This would be useful to determine whether DePO still offer improvement to the other methods (while using the same base model).
>
> **Reply:** Thank you for raising this point regarding model diversity. **We have extended our study to include an architecturally different backbone, Llama-3.1-8B Instruct, using the same TOMG-Bench setup as in Tab. 1. Notably, DePO improves performance on this architecturally distinct 8B backbone.**
>
> Specifically, we replace Qwen-2.5-3B Instruct with Llama-3.1-8B Instruct, which differs substantially from Qwen-2.5 in both architecture and tokenizer, and train the model on the TOMG-Bench property optimization tasks with the same baselines (SFT, GRPO, and GRPO with SFT initialization). As shown below, DePO consistently outperforms the backbone and all baselines:
>
> | LogP | SuccessRate $\uparrow$ | Sim $\uparrow$ | SuccessRate $\times$ Sim $\uparrow$ |
> | - | - | - | - |
> | Baseline (Llama-3.1-8B Instruct) | 0.237 | 0.692 | 0.164 |
> | SFT | 0.287 | 0.763 | 0.219 |
> | GRPO | 0.197 | 0.763 | 0.151 |
> | GRPO (SFT_Init) | 0.192 | 0.775 | 0.149 |
> | **DePO** | **0.360** | 0.746 | **0.269** |
>
> | QED | SuccessRate $\uparrow$ | Sim $\uparrow$ | SuccessRate $\times$ Sim $\uparrow$ |
> | - | - | - | - |
> | Baseline (Llama-3.1-8B Instruct) | 0.159 | 0.722 | 0.115 |
> | SFT | 0.185 | 0.809 | 0.150 |
> | GRPO | 0.115 | 0.805 | 0.093 |
> | GRPO (SFT_Init) | 0.121 | 0.811 | 0.098 |
> | **DePO** | **0.243** | 0.783 | **0.190** |
>
> | MR | SuccessRate $\uparrow$ | Sim $\uparrow$ | SuccessRate $\times$ Sim $\uparrow$ |
> | - | - | - | - |
> | Baseline (Llama-3.1-8B Instruct) | 0.169 | 0.763 | 0.129 |
> | SFT | 0.230 | 0.808 | 0.186 |
> | GRPO | 0.142 | 0.819 | 0.117 |
> | GRPO (SFT_Init) | 0.141 | 0.827 | 0.116 |
> | **DePO** | **0.293** | 0.789 | **0.231** |
>
> **These results on Llama-3.1-8B Instruct reinforce DePO’s cross-backbone robustness.** Across all three properties, DePO attains the highest SuccessRate $\times$ Sim under Llama-3.1-8B Instruct, mirroring the trends observed with the Qwen-2.5 series. This indicates that DePO’s gains are not specific to a single model family.
>
> In summary, DePO’s training objective is model-agnostic, its improvements already scale within the Qwen family, and its cross-backbone effectiveness is supported by experiments on Llama-3.1-8B Instruct.
>
> We will incorporate these new results into the revised manuscript, including full TOMG-Bench metrics and comparisons against the same baselines (Tab. 1), and we will explicitly highlight that DePO remains beneficial when applied to diverse and stronger backbone models.

---

> ### Author Response · Authors · 2025-11-27
> **Response to Reviewer Aw6o (part 2/3)**
>
> **(1) Constraining exploration is a principled, task-aligned design for similarity-constrained molecular optimization.** The TOMG-Bench tasks explicitly define a similarity-constrained objective, requiring optimized molecules $m^{\ast}$ to satisfy $\text{Similarity}(m^{*}, m_{0}) \ge \delta$ as formalized in Eqn. 1. In this setting, unconstrained exploration can easily drift into low similarity or chemically unrealistic molecules that achieve high predicted property values while violating the task specification. DePO’s demonstration-guided term biases the policy toward chemotypes that are both property-improving and structurally related to $m_{0}$, which acts as a task-aligned regularizer rather than an arbitrary restriction.
>
> Empirically, the results in Tabs. 12 to 15 in the original submission show that DePO outperforms baseline approaches by a large margin, including both Success Rate and Similarity. This indicates that constraining exploration steers the policy into a promising region of the search space that satisfies the similarity constraint and improves optimization quality.
>
> **(2) The main disadvantage is for tasks that truly require large structural modifications.** The main disadvantage of a similarity-constrained policy arises in settings where the optimal molecules lie far from the starting structure, such as fully unconstrained de novo design or diversity-oriented molecular generation. In these regimes, a strong similarity regularizer can lead to overly conservative exploration and reduced scaffold diversity. A variant of DePO that uses a weaker similarity weight or gradually relaxes the constraint over time would be preferable in such scenarios. However, the tasks considered in this paper are explicitly designed for similarity-constrained optimization, where practitioners care about preserving the structure of $m_{0}$ while adjusting properties.
>
> **(3) Error ranges are modest and broadly similar across methods; DePO’s gains are larger than the corresponding standard deviations.** We report the mean $\pm$ standard deviation over multiple random seeds for the TOMG-Bench property optimization task, shown as follows:
>
> | LogP | Success Rate $\uparrow$ | Sim $\uparrow$ | Success Rate$\times$Sim $\uparrow$ |
> | - | - | - | - |
> | Baseline | 0.269 $\pm$ 0.022 | 0.628 $\pm$ 0.016 | 0.169 |
> | Distill-SFT | 0.246 $\pm$ 0.011 | 0.553 $\pm$ 0.023 | 0.136 |
> | SFT | 0.323 $\pm$ 0.026 | 0.700 $\pm$ 0.008 | 0.226 |
> | GRPO | 0.179 $\pm$ 0.174 | 0.817 $\pm$ 0.025 | 0.146 |
> | GRPO (SFT init) | 0.183 $\pm$ 0.026 | 0.897 $\pm$ 0.030 | 0.164 |
> | DePO | 0.381 $\pm$ 0.029 | 0.734 $\pm$ 0.016 | 0.280 |
>
> | QED | Success Rate $\uparrow$ | Sim $\uparrow$ | Success Rate$\times$Sim $\uparrow$ |
> | - | - | - | - |
> | Baseline | 0.196 $\pm$ 0.025 | 0.666 $\pm$ 0.039 | 0.131 |
> | Distill-SFT | 0.220 $\pm$ 0.010 | 0.585 $\pm$ 0.012 | 0.129 |
> | SFT | 0.317 $\pm$ 0.023 | 0.701 $\pm$ 0.004 | 0.222 |
> | GRPO | 0.087 $\pm$ 0.044 | 0.905 $\pm$ 0.015 | 0.079 |
> | GRPO (SFT init) | 0.169 $\pm$ 0.048 | 0.893 $\pm$ 0.026 | 0.151 |
> | DePO | 0.264 $\pm$ 0.042 | 0.797 $\pm$ 0.035 | 0.210 |
>
> | MR | Success Rate $\uparrow$ | Sim $\uparrow$ | Success Rate$\times$Sim $\uparrow$ |
> | - | - | - | - |
> | Baseline | 0.244 $\pm$ 0.007 | 0.687 $\pm$ 0.004 | 0.167 |
> | Distill-SFT | 0.225 $\pm$ 0.016 | 0.590 $\pm$ 0.026 | 0.132 |
> | SFT | 0.391 $\pm$ 0.023 | 0.679 $\pm$ 0.008 | 0.266 |
> | GRPO | 0.098 $\pm$ 0.012 | 0.880 $\pm$ 0.016 | 0.086 |
> | GRPO (SFT init) | 0.158 $\pm$ 0.005 | 0.902 $\pm$ 0.004 | 0.142 |
> | DePO | 0.384 $\pm$ 0.008 | 0.776 $\pm$ 0.005 | 0.298 |
>
> From the results above, the **error bars (standard deviations) are small and of similar size across methods**. At the same time, **DePO’s gains in $\text{SR} \times \text{Sim}$ are much larger than these errors**. For example, on MR, DePO achieves $0.298$ compared with SFT’s $0.266$, so the improvement is much bigger than the typical variation across seeds. We see the same pattern for LogP and QED, which shows that DePO does not make training more unstable and that its performance gains are more robust rather than due to noise.
>
> In summary, constraining the exploration space in DePO is a principled and empirically supported design choice for similarity-constrained molecular optimization. It aligns directly with the TOMG-Bench objective, prevents the policy from drifting to low similarity or unrealistic molecules, and yields higher performance than unconstrained baselines, with standard deviations that are comparable to or smaller than other methods.

---

> ### Author Response · Authors · 2025-11-27
> **Response to Reviewer Aw6o (part 3/3)**
>
> In the revision, we will (i) emphasize around Eqn. 1 that DePO is specifically designed for similarity-constrained optimization and discuss its limitations for tasks that require large structural changes, and (ii) include multi-seed tables with mean $\pm$ standard deviation.
>
> **We thank Reviewer Aw6o again** for the careful review and for considering our rebuttal. **If our responses have addressed your concerns, we would be very grateful if you could reconsider your overall rating.** Any further comments or questions you may have are welcome!

---

### Official Review · Reviewer_h5YJ · 2025-10-31

**Soundness:** 3
**Presentation:** 3
**Contribution:** 3
**Rating:** 6
**Confidence:** 2

**Summary:**

A solid and well-executed paper proposing Demonstration-guided Policy Optimization (DePO) for LLM-based molecular optimization. The idea of integrating demonstration-guided supervision into RLVR is clear, novel enough for ICLR, and empirically convincing. Methodologically clean, though somewhat incremental from GRPO + SFT.

**Strengths:**

1. Novel combination of demonstration and RL: Clear motivation and principled integration of supervised guidance into RL objective.

2. Strong empirical gains: Up to 13% improvement over SFT and GRPO on TOMG-Bench and MuMOInstruct, with convincing generalization and ablations.

3. Well-written and interpretable: The framework (gradient masking, demonstration term) is intuitive and illustrated clearly with chemical reasoning examples.

**Weaknesses:**

1. Incremental contribution: Conceptually close to existing RLHF/RLVR + imitation learning hybrids; novelty may feel limited.

2. Limited scope: Only tested on molecular optimization—no evidence DePO generalizes to other scientific or reasoning domains.

3. Weak theoretical insight: Lacks formal analysis of convergence or policy improvement guarantees under demonstration bias.

**Questions:**

1. How sensitive is DePO to low-quality or noisy demonstrations (e.g., suboptimal molecules)?

2. Could DePO degenerate into simple imitation when β is high? How is β chosen in practice?

3. Does DePO still help when using stronger base models (e.g., >7B) trained with rich CoT reasoning data?

---

> ### Author Response · Authors · 2025-11-27
> **Response to Reviewer h5YJ (part 1/5)**
>
> **We thank Reviewer h5YJ for the constructive feedback.** In this rebuttal, we **address all your concerns** about (i) the contribution of DePO compared to RLHF/RLVR-style hybrids, (ii) whether DePO generalizes beyond molecular optimization, (iii) its robustness to noisy or biased demonstrations and the risk of imitation collapse at large $\beta$, and (iv) the role of the regularization term and the reliability of our comparisons on stronger base models.
> - First, we clarify how DePO is explicitly derived from GRPO’s failure modes and how its demonstration-guided, reasoning-masked objective differs structurally from standard RLHF/RLVR hybrids.
> - Second, we provide new results on a mathematical reasoning benchmark (MATH) and ablations on noisy demonstrations and varying $\beta$, showing that DePO is both domain-agnostic and robust.
> - Third, we explain the GRPO-inherited KL regularizer and present additional experiments on larger backbones (Llama-3.1-8B Instruct), where DePO continues to yield consistent gains.
> Point-by-point responses are provided below, together with the corresponding revisions we will make in the manuscript.
>
> ### W1: About the contribution
> > Incremental contribution: Conceptually close to existing RLHF/RLVR + imitation learning hybrids; novelty may feel limited.
>
> **Reply:** Thank you for this comment. **We would like to clarify that DePO is explicitly motivated by failure modes of GRPO in our setting and is designed to address them, using a demonstration-guided objective and reasoning-aware masking that are not present in GRPO or standard RLHF-style hybrids.** The resulting gains are substantial in both molecule quality and reasoning quality, and the formulation is domain agnostic and does not require collecting chain-of-thought annotations.
>
> **(1) GRPO fails in this domain due to the lack of valid reasoning supervision and the difficulty of exploring chemical space.**  As analyzed in Sec. 3 and Fig. 2, GRPO starting from an SFT model with only question-answer supervision is unable to recover multi-step reasoning (Obs. 3.2). Its trajectories quickly lose structured reasoning and collapse to short, template-like responses. At the molecule level, GRPO also exhibits overly conservative exploration (Obs. 3.1), repeatedly producing molecules that are almost identical to the input because of sparse rewards and the similarity constraint in TOMG Bench. These behaviors indicate that in the absence of reliable priors or curated reasoning traces, GRPO alone cannot balance property improvement, diversity, and coherent reasoning. This diagnosis is the starting point for the DePO design and shows that the problem is not simply tuning GRPO but addressing its structural limitations.
>
> **(2) DePO introduces a demonstration-guided objective tailored to these failure modes, which differs from generic RLHF or RLVR-style hybrids.** Standard RLHF or RLVR style methods typically combine a reward term with a supervised log probability term on the same full trajectory. In our setting, however, we do not have ground truth reasoning sequences, only optimized molecules from TOMG Bench and MuMOInstruct. DePO therefore uses a demonstration term that conditions on the model-generated reasoning prefix $t_i$ but supervises only the final molecule $a_i^{\ast}$, with gradients masked on reasoning tokens. Concretely, the DePO loss adds a term of the form $\beta \cdot \log \pi_\theta(a_i^{\ast} \mid q, t_i)$ on top of the GRPO objective (Eqn. 4), while the RL advantage $\hat{A}_{i,k}$ continues to update all tokens in the reasoning and answer sequence based on property rewards. This combination anchors the policy in a region of RDKit-validated, property-improving molecules, while still allowing exploration through the GRPO update. The reasoning is that aware masking avoids backpropagating incorrect supervision through model-generated thoughts. This structure is specifically designed to solve the failure modes in Fig. 2 and is not equivalent to simply adding an imitation term on full trajectories as in conventional RLHF hybrids.
>
> **(3) The improvements provided by DePO are substantial, consistent, and qualitatively distinct from GRPO.** Across all TOMG-Bench tasks, DePO achieves clear gains over GRPO (up to about 13 percentage points in Tab. 1), while maintaining or improving similarity to the original molecules. These gains persist in MuMOInstruct (Tab. 2), indicating that DePO’s effect is robust rather than a narrow tweak. Our additional LLM-as-judge evaluation as follows confirms that DePO’s trajectories receive higher reasoning quality scores than GRPO under the same base model and training setup.

---

> ### Author Response · Authors · 2025-11-27
> **Response to Reviewer h5YJ (part 2/5)**
>
> Specifically, we introduce a quantitative evaluation of reasoning quality using an LLM-as-judge protocol calibrated to expert chemists’ assessments [1]. For TOMG-Bench LogP optimization, we sample 50 trajectories per method and score the reasoning quality of each explanation. The results are shown below:
>
> | Method | Score $\uparrow$ |
> | - | - |
> | Baseline | 4.16 $\pm$ 1.32 |
> | GRPO | 3.60 $\pm$ 1.39 |
> | **DePO (Ours)** | **4.32 $\pm$ 1.63** |
>
> Together, these results indicate that DePO changes both what the model optimizes and how it reasons, beyond what would be expected from a minor objective variation.
>
> In summary, DePO is a principled extension to GRPO that is directly derived from the observed weaknesses of GRPO in chemical property optimization: it adds a demonstration-guided term and reasoning-aware masking that address collapsed exploration and degraded reasoning in a way that standard RLHF- or RLVR-style hybrids do not. The improvements in Tabs. 1-2 and the qualitative and quantitative reasoning analyses (the LLM-as-judge results) show that the impact is significant rather than incremental.
>
> In the revision, we will (i) emphasize the GRPO failure analysis in Sec. 3 as the motivation for DePO and (ii) clarify how DePO’s objective structurally differs from existing RLHF-style hybrids.
>
> Reference:
> [1] Reasoning-Enhanced Large Language Models for Molecular Property Prediction. In arXiv, 2025.
>
> ### W2: Extend DePO to task beyond molecular optimization
> > Limited scope: Only tested on molecular optimization—no evidence DePO generalizes to other scientific or reasoning domains.
>
> **Reply:** Thank you for this comment. **We would like to clarify that DePO is designed for improving exploration under sparse, verifiable rewards, and we have conducted new experiments on a mathematical reasoning task (MATH) to substantiate this claim.** These results show that DePO improves the model's performance beyond chemistry.
>
> **(1) DePO is formulated in a domain-agnostic way and does not rely on chemistry-specific components.** The DePO objective combines a sparse verifiable reward and a demonstration-guided term on final outputs. It does not encode domain-specific rules such as RDKit heuristics into the model or loss. The only requirements are (i) a reward function that can verify final outputs and (ii) a set of demonstration outputs that are known to be high quality. This is the same setup in molecular optimization, where the reward is a property score and demonstrations are optimized molecules, and it naturally extends to other domains where final answers are verifiable.
>
> **(2) DePO improves reasoning beyond the chemical domain.** To test generalization, we apply DePO to the MATH benchmark [1], a math reasoning task where the reward is binary ($r = 1$ if the final answer is correct and $r = 0$ otherwise). We treat ground truth solutions as demonstrations and adopt the same decomposition $o_i = [t_i; \hat{a}_i]$ as in Fig. 4, where $t_i$ is the model-generated chain of thought and $\hat{a}_i$ is the final answer. Using Qwen-2.5-3B Instruct as the base model, we train GRPO and DePO under the same setup and test them on the MATH test set.
>
> | MATH   | Accuracy $\uparrow$ |
> | - | - |
> | Baseline | 0.598 |
> | GRPO | 0.600 |
> | **DePO** | **0.603** |
>
> As shown in the table above, DePO achieves higher final answer accuracy than GRPO, showing that the demonstration-guided objective remains beneficial when the structure, domain knowledge, and data format are entirely different from chemistry.
>
> In summary, the new MATH experiments demonstrate that DePO yields consistent gains in a distinct reasoning domain, improving answer accuracy over GRPO and the baseline.
>
> In the revision, we will (i) include these MATH results in a new subsection, (ii) report the corresponding accuracy, and (iii) expand the discussion to emphasize that DePO is a general framework for demonstration-guided optimization under sparse verifiable rewards, not limited to molecular tasks.
>
> References:
> [1] Measuring mathematical problem solving with the math dataset. In NeurIPS, 2021.

---

> ### Author Response · Authors · 2025-11-27
> **Response to Reviewer h5YJ (part 3/5)**
>
> ### W3&Q1: DePO’s robustness on noisy demonstrations
> > Weak theoretical insight: Lacks formal analysis of convergence or policy improvement guarantees under demonstration bias.
> > How sensitive is DePO to low-quality or noisy demonstrations (e.g., suboptimal molecules)?
>
> **Reply:** Thank you for raising this question. **In short, DePO benefits from high-quality demonstrations, but it is not brittle: performance degrades smoothly as demonstrations are corrupted and, in the worst case, reverts to baseline performance rather than collapsing.** We support this claim with two complementary robustness experiments.
>
> **(1) DePO maintains strong gains under moderate noise and reverts toward baseline under extreme corruption.** We simulate noise at the subtask level on the TOMG-Bench property optimization task. Each benchmark consists of subtasks that share the same optimization target. For each subtask, we take its list of demonstration molecules and **select each demonstration with probability (p); the selected molecules are then randomly permuted within that subtask**. This keeps the per-subtask molecule distribution fixed but deliberately breaks the alignment between individual queries and their demonstrations. We then train DePO on the corrupted datasets using Qwen-2.5-3B Instruct and report Success Rate and Similarity.
>
> | LogP | Success Rate $\uparrow$ | Sim $\uparrow$ | Success Rate $\times$ Sim $\uparrow$ |
> | - | - | - | - |
> | Baseline (Qwen 2.5-3B Instruct) | 0.268 | 0.627 | 0.168 |
> | DePO (50 % corrupted) | 0.290 | 0.616 | 0.179 |
> | DePO (30 % corrupted) | 0.283 | 0.686 | 0.194 |
> | **DePO (full)** | **0.415** | **0.715** | **0.297** |
>
> | MR | SuccessRate $\uparrow$ | Sim $\uparrow$ | SuccessRate $\times$ Sim $\uparrow$ |
> | - | - | - | - |
> | Baseline (Qwen 2.5-3B Instruct) | 0.252 | 0.685 | 0.173 |
> | DePO (50 % corrupted) | 0.248 | 0.687 | 0.170 |
> | DePO (30 % corrupted) | 0.289 | 0.714 | 0.206 |
> | **DePO (full)** | **0.399** | **0.736** | **0.294** |
>
> | QED | SuccessRate $\uparrow$ | Sim $\uparrow$ | SuccessRate $\times$ Sim $\uparrow$ |
> | - | - | - | - |
> | Baseline (Qwen 2.5-3B Instruct) | 0.188 | 0.693 | 0.130 |
> | DePO (50 % corrupted) | 0.198 | 0.706 | 0.140 |
> | DePO (30 % corrupted) | 0.228 | 0.728 | 0.166 |
> | **DePO (Ours)** | **0.312** | **0.756** | **0.236** |
>
> From the results shown above,
> - with 30 % corrupted demonstrations, DePO still yields **clear improvements over the baseline** across all three properties, indicating that DePO continues to extract useful signal even when a non-trivial fraction of demonstrations is misaligned.
> - with 50 % corrupted demonstrations, DePO’s performances are still **competitive** with the baseline on all three properties. This shows that when demonstrations are heavily disrupted at the subtask level, DePO behaves comparably to running the backbone model alone rather than causing catastrophic degradation.
>
> Taken together, these results show that high-quality demonstrations yield substantial gains, moderate corruption still brings noticeable improvements, and extreme corruption primarily removes the benefit of the demonstration term instead of destabilizing training.
>
> **(2) Dropping demonstrations entirely leads to similarly graceful degradation.** We also study a complementary setting where demonstrations are *missing* rather than misassigned. In Appendix E.3 (Tab. 9), we randomly drop the demo-guided loss for a subset of training examples (i.e., only **40\%** of demonstrations are used). Even in this pessimistic setting, DePO with only 40\% of demos still **outperforms** the baseline on all three properties and remains close to full DePO, confirming that DePO does not require dense, perfectly complete demonstrations to be effective.
>
> In summary, DePO is not fragile: performance degrades smoothly and returns to the regime of the backbone model rather than diverging or collapsing.
>
> In the revision We will add the “subtask-level corruption” ablations to the appendix to show the DePO’s behavior under noisy or partially unreliable demonstrations is stable and degrades gracefully instead of failing unpredictably.

---

> ### Author Response · Authors · 2025-11-27
> **Response to Reviewer h5YJ (part 4/5)**
>
> ### Q2: DePO does not degenerate into imitation under various choices of $\beta$
> > Could DePO degenerate into simple imitation when β is high? How is β chosen in practice?
>
> **Reply:** Thank you for this thoughtful question. In brief, DePO does **not** collapse into simple imitation even for large $\beta$ (from $1.0$ to $100.0$), because the demonstration term is applied only to final answer tokens while the thinking tokens are always optimized by the RL objective, and empirically, DePO is stable across a wide range of $\beta$ values. In practice, we simply set $\beta = 1.0$ based on a coarse ablation (Fig. 5) and do not tune it per task.
>
> **(1) Gradient masking ensures that demonstrations supervise only the final answer, not the reasoning trajectory.** DePO explicitly decomposes each trajectory as $o_i = [t_i; \hat{a_i}]$, where $t_i$ denotes the reasoning tokens and $\hat{a_i}$ denotes the final answer tokens aligned with the demonstrated answer $a_i$. As described in Sec. 3 and illustrated in Fig. 4, the demonstration loss is applied **only** to $\hat{a_i}$, while the gradients on $t_i$ from the demonstration term are masked out. In Eqn. 4, the Exploration term is the RL objective over **all** tokens, the Guidance term is the demonstration-guided loss **only on $\hat{a}_i$** (replaced by $a_i$), and the KL term stabilizes updates. The gradient of the second term is zero on $t_i$ by construction. As a result, even for large $\beta$, the model is never trained to imitate the reasoning sequence of the demonstrations. The reasoning tokens remain driven by the reward signal through the advantage $\hat{A}_{i,k}$, which preserves exploration and prevents collapse into behavior cloning.
>
> **(2) Increasing $\beta$ strengthens answer guidance but does not turn DePO into behavior cloning.** From Eq. (4), increasing $\beta$ only scales the supervised signal on the final answer tokens. It does **not** introduce any supervised loss on the intermediate reasoning tokens and does **not** remove the RL term. This means that, even when $\beta$ is large, DePO still optimizes the reward over the entire trajectory and uses the demonstration only as a preference on which final answers are desirable. In particular, the policy $\pi_\theta$ is encouraged to reach answer regions consistent with $a_i$, but it is free to discover different reasoning paths $t_i$ that achieve higher reward. This structural separation between answer supervision and trajectory-level RL is precisely what prevents DePO from degenerating into simple imitation learning.
>
> **(3) Empirically, DePO is robust across a wide range of $\beta$ and does not converge to SFT behavior.** Fig. 5 provides a direct ablation over $\beta \in [0.0, 1.0, 10.0, 100.0]$. The results show that DePO maintains strong performance for moderate $\beta$ and exhibits only mild degradation at very large $\beta$. Importantly, the performance curve does **not** converge to the SFT baseline as $\beta$ increases. Instead, DePO with $\beta = 10.0$ and $\beta = 100.0$ still remains clearly distinct from SFT and retains the benefits of RL optimization. This empirical pattern confirms that increasing $\beta$ does not cause DePO to behave like pure imitation; rather, it trades off the strength of answer guidance while maintaining a reward-driven update on the reasoning trajectory.
>
> **(4) In practice, we choose $\beta$ once and keep it fixed across tasks.** In our experiments, we perform a coarse sweep over $\beta$ using Fig. 5 and observe that performance is relatively flat in a broad range around $\beta = 1.0$. Based on this observation, we adopt a simple and practical choice: we set $\beta = 1.0$ as a default for all benchmarks without further per-task tuning. This reflects that DePO is not highly sensitive to the exact value of $\beta$ and that a single global setting suffices in practice.
>
> In summary, DePO does not degenerate into simple imitation because the demonstration signal is structurally restricted to final answer tokens while the reasoning trajectory is always optimized via the RL objective. The form of Eqn. 4, together with the gradient masking mechanism in Sec. 4.2 and the ablation in Fig. 5, jointly show that exploration is preserved and that DePO is robust to a wide range of $\beta$ values.
>
> In the revision, we will (i) make this decomposition explicit when presenting Eqn. 4, (ii) add an explanation of how gradient masking prevents imitation collapse in Sec. 3, and (iii) explicitly reference Fig. 5 when discussing the practical choice of $\beta$.

---

> ### Author Response · Authors · 2025-11-27
> **Response to Reviewer h5YJ (part 5/5)**
>
> ### Q3: Effectiveness on stronger base models
> > Does DePO still help when using stronger base models (e.g., >7B) trained with rich CoT reasoning data?
>
> **Reply:** Thank you for raising this important point regarding the applicability of DePO to larger backbones. In brief, DePO continues to provide consistent gains on a larger and architecturally different base model, **Llama-3.1-8B Instruct**, under the same TOMG-Bench settings.
>
> **DePO improves performance on a larger 8B instruction model with a different architecture.** To evaluate cross-backbone robustness, we adopt **Llama-3.1-8B Instruct**, which differs substantially in architecture and tokenizer, and train the model using the TOMG-Bench property optimization task using the same protocol as Tab. 1. As shown below, DePO consistently outperforms the backbone and all fine-tuning baselines (SFT, GRPO, and GRPO with SFT initialization) in terms of Success Rate, similarity, and their product (Success Rate $\times$ Sim) across LogP, QED, and MR:
>
> | LogP | SuccessRate $\uparrow$ | Sim $\uparrow$ | SuccessRate $\times$ Sim $\uparrow$ |
> | - | - | - | - |
> | Baseline (Llama 3.1 8B Instruct) | 0.237 | 0.692 | 0.164 |
> | SFT | 0.287 | 0.763 | 0.219 |
> | GRPO | 0.197 | 0.763 | 0.151 |
> | GRPO (SFT_Init) | 0.192 | 0.775 | 0.149 |
> | **DePO** | **0.360** | 0.746 | **0.269** |
>
> | QED | SuccessRate $\uparrow$ | Sim $\uparrow$ | SuccessRate $\times$ Sim $\uparrow$ |
> | - | - | - | - |
> | Baseline (Llama 3.1 8B Instruct) | 0.159 | 0.722 | 0.115 |
> | SFT | 0.185 | 0.809 | 0.150 |
> | GRPO | 0.115 | 0.805 | 0.093 |
> | GRPO (SFT_Init) | 0.121 | 0.811 | 0.098 |
> | **DePO** | **0.243** | 0.783 | **0.190** |
>
> | MR | SuccessRate $\uparrow$ | Sim $\uparrow$ | SuccessRate $\times$ Sim $\uparrow$ |
> | - | - | - | - |
> | Baseline (Llama 3.1 8B Instruct) | 0.169 | 0.763 | 0.129 |
> | SFT | 0.230 | 0.808 | 0.186 |
> | GRPO | 0.142 | 0.819 | 0.117 |
> | GRPO (SFT_Init) | 0.141 | 0.827 | 0.116 |
> | **DePO** | **0.293** | 0.789 | **0.231** |
>
> These results show that, even when starting from a larger 8B instruction model, **DePO preserves the same qualitative behavior as in Tab. 1**: it improves optimization success while maintaining high similarity, leading to the best SuccessRate $\times$ Sim among all methods.
>
> The results indicate that **plain GRPO brings little to no gain over the already strong base model, whereas DePO yields clear improvements across all three properties**. This suggests that DePO effectively leverages the base model’s reasoning ability while steering it toward property-aware, similarity-constrained exploration that is better aligned with molecular optimization objectives and less prone to repetitive responses.
>
> In summary, DePO consistently improves optimization performance on stronger and architecturally diverse base models, which were trained with rich CoT reasoning data.
>
> In the revision, we will include these new results in the revision and explicitly highlight that DePO is compatible with and beneficial for larger backbones, rather than being limited to smaller models.
>
> **We thank Reviewer h5YJ again** for the careful review and for considering our rebuttal. **If our responses have addressed your concerns, we would be very grateful if you could reconsider your overall rating.** Any further comments or questions you may have are welcome!

---

### Official Review · Reviewer_knP9 · 2025-10-31

**Soundness:** 3
**Presentation:** 3
**Contribution:** 2
**Rating:** 4
**Confidence:** 4

**Summary:**

The paper introduces DePO a framework that enhances large language models’ reasoning ability for molecular optimization by combining reinforcement learning with expert demonstrations. Unlike prior approaches such as SFT, which suppress reasoning, or RL with verifiable rewards, which suffers from unguided exploration, DePO integrates reference molecules as supervised signals to guide the search toward chemically valid regions while preserving intermediate reasoning steps through gradient masking. Evaluated on TOMG-Bench and MuMOInstruct, DePO achieves up to 13% performance gains over strong baselines, demonstrating improved balance between property optimization and structural similarity, better generalization to unseen instructions, and chemically coherent reasoning, establishing a new paradigm for LLM-driven scientific reasoning and molecular design.

**Strengths:**

The paper introduces a novel demonstration-guided reinforcement learning paradigm (DePO) that effectively bridges the gap between language-model reasoning and domain-constrained scientific optimization. It presents comprehensive experiments across multiple molecular optimization benchmarks, providing strong empirical evidence for the framework’s effectiveness. The significance of this work lies in extending LLM reasoning beyond text and mathematics to molecular design, demonstrating that demonstration-guided optimization can deliver substantial improvements in both performance and generalization within complex chemistry-related tasks.

**Weaknesses:**

1. The discussion of related work is limited. Several recent GPT-based molecular optimization studies are neither cited nor compared as baselines, making it difficult to contextualize DePO’s contributions within existing literature.

2. The novelty appears somewhat incremental, as the method primarily adds guiding exploration and a regularization term to the GRPO objective. The introduced regularization is heuristic in nature, and the paper does not provide sufficient discussion on how the hyperparameters (β and γ) are selected or tuned, which may significantly influence final performance.

3. Although the paper motivates DePO’s gradient-masking mechanism, it lacks theoretical justification or a detailed ablation study to assess how different masking strategies impact reasoning fidelity, convergence, and optimization stability.

4. While DePO presents qualitative improvements in reasoning traces, the evaluation of reasoning quality remains largely subjective. Incorporating quantitative reasoning metrics or expert chemical validation would make the claims more convincing.

5. how to create the demonstration dataset is not discussed.

**Questions:**

Additional questions regarding weaknesses:

1. The design and formulation of the regularization component are not clearly explained.

2. Why is the gradient flow disabled during the reasoning (“thinking”) phase? Could the authors provide an ablation study to justify this choice?

3. In Figure 7, there is a large discrepancy between DePO and GRPO. The result appears somewhat unreliable, as GRPO seems to lack a valid expression. Were both methods fine-tuned on the same base model?
Without providing the source code, it's hard to believe such difference considering on overall similarity in objective function.

---

> ### Author Response · Authors · 2025-11-27
> **Response to Reviewer knP9 (part 1/7)**
>
> **We thank Reviewer knP9 for the constructive and detailed feedback.** In this rebuttal, we address your concerns about (i) related work and GPT-based molecular optimization, (ii) the novelty and hyperparameters of DePO, (iii) masking strategies and reasoning evaluation, (iv) construction of the demonstration dataset, (v) the regularization term, and (vi) the reliability of the DePO–GRPO comparison.
>
> - First, we move and expand the discussion of GPT-based molecular optimization into the main related-work section and add GPT-based and foundation-model baselines on TOMG-Bench, where DePO remains competitive or superior.
> - Second, we clarify how DePO’s demonstration guidance and gradient masking directly address GRPO’s failure modes and show that performance is robust to $\beta$, while $\gamma$ is inherited unchanged from GRPO.
> - Third, we add ablations over masking strategies, an LLM-as-judge evaluation, and training-dynamics analysis, showing that structured masking improves both optimization stability and reasoning quality.
> - Fourth, we explicitly describe how demonstrations are automatically derived from benchmark ground-truth optimized molecules and clarify that they consist of instruction-output pairs without reasoning traces.
> - Finally, we explain that the regularization term is the standard PPO-style KL penalty reused from GRPO and confirm that DePO and GRPO share the same base model and training pipeline, with quantitative evidence supporting the qualitative gap in Fig. 7.
>
> Point-by-point responses are provided below, together with the concrete revisions we will make in the manuscript.
>
> ### W1: Discussing and comparing with GPT-based molecular optimization methods
> > The discussion of related work is limited. Several recent GPT-based molecular optimization studies are neither cited nor compared as baselines, making it difficult to contextualize DePO’s contributions within existing literature.
>
> **Reply:** Thank you for this comment. In response, we (i) **clarify that GPT-based molecular optimization methods are already discussed in Appendix E.1** and promote and expand this discussion into the main related work section, (ii) **position the uniqueness of DePO compared to existing GPT-based optimizers**, and (iii) **conduct additional experiments with GPT-based baselines on TOMG Bench**, where DePO achieves competitive or superior performance while using the same backbone.
>
> **(1) GPT-based molecular optimization methods are already analyzed in Appendix E.1, and this discussion will be promoted into the main related work.** In the original submission, we devoted Appendix E.1 to a focused discussion of GPT and LLM-based molecular optimization methods, including approaches that use LLMs to guide mutation or crossover and those that adapt molecular foundation models.  In the revision, we will move this discussion into the main related-work section and expand it with additional recent GPT-based studies. We explicitly distinguish three families:
>
> - **Black-box GPT optimizers** (e.g., MOLLEO [1], LLM-guided genetic or evolutionary algorithms): LLMs are used as proposal generators inside external optimizers (crossover/mutation operators, scoring heuristics), but the optimization logic remains outside the LLM. The model is not trained end-to-end to reason about trade-offs under similarity constraints.
> - **Property-prediction LLMs** (e.g., LICO-style in-context optimizers [2]): LLMs are fine-tuned or prompted as surrogate property predictors that plug into a Bayesian or other search loop. They learn to predict properties of molecules but not how to modify molecules to jointly satisfy property and similarity constraints.
> - **Domain-specific foundation models** (e.g., BioT5 [3], MolT5 [4]): these models pretrain on large chemical corpora but are not explicitly trained for stepwise, instruction-following optimization with competing objectives.
>
> Notably, **DePO is positioned as a reasoning-centric, end-to-end optimizer**: it directly trains the LLM as the optimization policy, not as a black-box helper. DePO integrates **supervised guidance** (using reference molecules), **gradient-masked RLVR**, and **explicit structural similarity constraints** to (i) maintain chain-of-thought style reasoning, (ii) explore the chemical space under task-specific constraints, and (iii) produce both the optimized molecule and its intermediate reasoning in a single policy. This combination is not present in prior GPT-based molecular optimization methods.
>
> **(2) We conduct experiments with GPT-based molecular optimizers and foundation models as baselines with matched backbones, and DePO remains competitive or superior.** We extend our experiments to include GPT-based molecular optimization methods and foundation models as baselines on TOMG-Bench property optimization (LogP, MR, QED):

---

> ### Author Response · Authors · 2025-11-27
> **Response to Reviewer knP9 (part 2/7)**
>
> - We employ MOLLEO, which integrates a pretrained LLM into a Graph-GA framework to guide crossover and mutation, and evaluate it under the same TOMG-Bench protocol. For fair comparison, we instantiate MOLLEO using the same backbone (Qwen-2.5-3B Instruct) as DePO, so any performance difference reflects the optimization framework rather than model capacity.
>
> - We further include classic molecular optimization baselines (Graph-GA [5], REINVENT [6]) and molecular foundation models (BioT5-base, MolT5-large), as well as general-purpose models (GPT-4o-mini), evaluated zero-shot on the same tasks. This yields a comprehensive comparison spanning graph-based, RL-based, and GPT-based approaches.
>
> Across LogP, MR, and QED, DePO achieves **higher or comparable Success Rate $\times$ Similarity** compared to all GPT-based baselines, while using the same or smaller backbones. Below we show the summary tables:
>
> | LogP | **Success Rate $\uparrow$** | **Sim. $\uparrow$** | **Success Rate $\times$ Sim. $\uparrow$** |
> | - | - | - | - |
> | Graph GA | 0.509 | 0.125 | 0.064 |
> | REINVENT | 0.465 | 0.125 | 0.058 |
> | MOLLO | 0.510 | 0.103 | 0.053 |
> | GPT-4o-mini | 0.499 | 0.706 | 0.352 |
> | Mol-T5-large | 0.424 | 0.102 | 0.043 |
> | Bio-T5-base | 0.516 | 0.153 | 0.079 |
> | Baseline (Qwen 2.5-3B Instruct) | 0.268 | 0.627 | 0.168 |
> | SFT | 0.298 | 0.692 | 0.206 |
> | GRPO | 0.379 | 0.806 | 0.305 |
> | GRPO (SFT-Init) | 0.212 | 0.863 | 0.183 |
> | **DePO (Ours)** | **0.415** | **0.715** | **0.297** |
>
> | MR | **Success Rate $\uparrow$** | **Sim. $\uparrow$** | **Success Rate $\times$ Sim. $\uparrow$** |
> | - | - | - | - |
> | Graph GA | 0.509 | 0.122 | 0.062 |
> | REINVENT | 0.595 | 0.116 | 0.069 |
> | MOLLO | 0.509 | 0.127 | 0.065 |
> | GPT-4o-mini | 0.409 | 0.771 | 0.315 |
> | Mol-T5-large | 0.450 | 0.107 | 0.048 |
> | Bio-T5-base | 0.506 | 0.160 | 0.081 |
> | Baseline (Qwen 2.5-3B Instruct) | 0.252 | 0.685 | 0.173 |
> | SFT | 0.359 | 0.663 | 0.238 |
> | GRPO | 0.214 | 0.880 | 0.188 |
> | GRPO (SFT-Init) | 0.265 | 0.850 | 0.225 |
> | **DePO (Ours)** | **0.399** | **0.736** | **0.294** |
>
> | QED | **Success Rate $\uparrow$** | **Sim. $\uparrow$** | **Success Rate $\times$ Sim. $\uparrow$** |
> | - | - | - | - |
> | Graph GA | 0.493 | 0.140 | 0.069 |
> | REINVENT | 0.558 | 0.115 | 0.064 |
> | MOLLO | 0.496 | 0.122 | 0.061 |
> | GPT-4o-mini | 0.231 | 0.752 | 0.174 |
> | Mol-T5-large | 0.465 | 0.119 | 0.055 |
> | Bio-T5-base | 0.507 | 0.158 | 0.080 |
> | Baseline (Qwen 2.5-3B Instruct) | 0.188 | 0.693 | 0.130 |
> | SFT | 0.297 | 0.697 | 0.207 |
> | GRPO | 0.138 | 0.889 | 0.123 |
> | GRPO (SFT-Init) | 0.223 | 0.863 | 0.193 |
> | **DePO (Ours)** | **0.312** | **0.756** | **0.236** |
>
> These results show that DePO’s **demonstration-guided RLVR with masked reasoning** is competitive and often superior to existing GPT-based molecular optimization strategies. DePO therefore provides an orthogonal, reasoning-focused design that can complement black-box GPT optimizers and domain-specific molecular LMs.
>
> In summary, DePO differs from these works by training the LLM as a reasoning-centric, end-to-end constrained optimization policy that combines demonstration guidance, gradient-masked RL, and similarity-aware rewards, rather than using the LLM only as a proposal generator or property predictor. The additional experiments further show that DePO achieves competitive or superior performance compared to other molecular optimization approaches.
>
> In the revised paper, we will (i) move the discussion on existing GPT-based methods from Appendix E.1 to the main related work section, (ii) add a dedicated paragraph that clearly states DePO’s uniqueness relative to prior GPT-based optimizers, and (iii) incorporate GPT-based baselines with matched backbones on TOMG Bench, reporting the full comparison tables in the main experiments and appendix.
>
> Reference:
> [1] Efficient Evolutionary Search Over Chemical Space with Large Language Models. In ICLR, 2025.
> [2] Lico: Large language models for in-context molecular optimization. In ICLR, 2025.
> [3] Biot5: Enriching cross-modal integration in biology with chemical knowledge and natural language associations. In EMNLP, 2023.
> [4] Translation between molecules and natural language. In EMNLP, 2022.
> [5] A graph-based genetic algorithm and generative model/Monte Carlo tree search for the exploration of chemical space. In Chemical Science, 2019.
> [6] Molecular de-novo design through deep reinforcement learning. In Journal of Cheminformatics, 2017.

---

> ### Author Response · Authors · 2025-11-27
> **Response to Reviewer knP9 (part 3/7)**
>
> ### W2: Clarifying DePO’s contributions and hyperparameter choices
> > The novelty appears somewhat incremental, as the method primarily adds guiding exploration and a regularization term to the GRPO objective. The introduced regularization is heuristic in nature, and the paper does not provide sufficient discussion on how the hyperparameters (β and γ) are selected or tuned, which may significantly influence final performance.
>
> **Reply:** Thank you for this comment. To clarify, **DePO is designed to address two concrete failure modes of GRPO** in similarity-constrained molecular optimization: (i) GRPO cannot balance the competitive molecular optimization constraints, and (ii) GRPO cannot recover the reasoning ability from the SFT-initialized model. To solve this, **DePO introduces a targeted solution combining demonstration-guided anchoring with a new gradient-masking mechanism**, which prevents conflicting supervision on intermediate reasoning tokens while allowing reward-driven optimization to shape the reasoning process.
>
> **(1) DePO directly targets two specific GRPO failure modes.** Sec. 3 and Fig. 2 show that vanilla GRPO, even with an SFT initialization, exhibits:
> - **Collapsed exploration.** GRPO tends to generate molecules that are nearly identical to the input scaffold (Obs. 3.1), which yields high similarity but limited property improvement.
> - **Failure to recover multi-step reasoning.** GRPO with SFT initialization often jumps directly from the question to an answer without meaningful intermediate reasoning (Obs. 3.2).
>
> These are not minor cosmetic issues; they are structural limitations that degrade optimization quality in the molecular setting that DePO aims to address.
>
> **(2) DePO’s gradient mask and demonstration-guided objective resolve the failure modes of GRPO.** DePO introduces two coupled components:
> - **Demonstration guidance for exploration.** Using dataset-derived demonstrations, the policy is nudged toward RDKit-validated, property-improving molecules that respect $\text{Similarity}(m^{*}, m_{0}) \ge \delta$, as described in Appendix E.6. This provides **anchoring signals** to escape GRPO’s near-identity solutions and discover nontrivial modifications while still obeying the similarity constraint. The RL objective remains unchanged and continues to drive fine-grained optimization around these demonstrations.
> - **Gradient masking on “thinking” tokens.** Instead of naively supervising the entire reasoning trajectory, DePO applies the supervised demonstration loss only to answer tokens and masks all intermediate “thinking” tokens, while the GRPO reward still updates the full trajectory.
>
> This design **avoids conflicting supervision** when the model’s internally generated reasoning does not match the demonstration. Without masking, teacher forcing on all tokens can push incorrect chains of thought toward the final answer, while the RL signal tries to improve reasoning based on reward, leading to gradient conflict and degraded trajectories. **Fig. 11 and Tab. 3 show that DePO with full gradient masking achieves better molecular optimization metrics and higher LLM-as-judge reasoning scores than both the “no mask” and random masking variants.** We kindly refer the reviewer to the responses in **W3&W4&Q2** for more details.
>
> **(3) DePO’s hyperparameters are stable and do not drive the observed improvements.** The weight $\beta$ is the **only** method-specific coefficient introduced by DePO. Fig. 5 reports an ablation over a wide range of $\beta$ values, demonstrating that DePO maintains consistent performance across settings, which confirms that the method is robust rather than sensitive to tuning. The coefficient $\gamma$ is inherited directly from GRPO’s PPO-style KL penalty and is not introduced by DePO. As in prior PPO and GRPO work, $\gamma$ serves only to stabilize policy updates and does not control exploration behavior, so we retain the same value for fairness.
>
> In summary, DePO introduces a necessary and principled correction to GRPO’s documented limitations, and its performance is not dependent on sensitive hyperparameter choices. We will revise the manuscript to clarify the motivation behind each component, explicitly distinguish between GRPO-inherited and DePO-specific hyperparameters, and emphasize the robustness results presented in Fig. 5.

---

> ### Author Response · Authors · 2025-11-27
> **Response to Reviewer knP9 (part 4/7)**
>
> ### W3&W4&Q2: Ablation on alternative masking strategies and quantitative evaluation of reasoning quality
> > Although the paper motivates DePO’s gradient-masking mechanism, it lacks theoretical justification or a detailed ablation study to assess how different masking strategies impact reasoning fidelity, convergence, and optimization stability.
> > While DePO presents qualitative improvements in reasoning traces, the evaluation of reasoning quality remains largely subjective. Incorporating quantitative reasoning metrics or expert chemical validation would make the claims more convincing.
> > Why is the gradient flow disabled during the reasoning (“thinking”) phase? Could the authors provide an ablation study to justify this choice?
>
> **Reply:** Thank you for these valuable suggestions. We would like to clarify that **gradient masking in DePO is applied only to the supervised (demonstration) loss on “thinking” tokens, while the RL objective still updates all reasoning tokens.** This masking is a central design choice in DePO, and its effect is already examined via the “no mask” ablation in Tab. 3. Building on that analysis, we further evaluate alternative masking strategies, quantitative reasoning scores, and training dynamics. Our main conclusion is that **masking all thinking tokens in the supervised term improves both molecular optimization performance and the quality of intermediate reasoning (as measured by LLM-as-Judge) compared with random masking and the no-mask variant.**
>
> **(1) Gradient masking prevents the supervised loss from incorrectly training on model-generated reasoning tokens.** In DePO, demonstrations provide only the final optimized molecule $a_i^{\ast}$ for a query $q$. The supervised term takes the form $\log \pi_\theta(a_i^{\ast} \mid q, t_i)$, where $t_i$ is the reasoning prefix generated by the current policy. Crucially, $t_i$ is not part of the demonstration ground truth. If we allow gradients to flow through $t_i$, the model is trained as if its own sampled reasoning tokens were the correct explanation for $a_i^{\ast}$. This misaligns credit assignment, because the same final molecule could be reached via many different, possibly incorrect, reasoning paths. As a result, the model would be encouraged to reinforce arbitrary or flawed reasoning patterns simply because they happened to precede a correct molecule. Masking gradients on $t_i$ ensures that the supervised signal is applied only where labels are valid, namely on the answer tokens, and avoids injecting spurious causal links between arbitrary reasoning and correct outcomes.
>
> **(2) Random-masking ablations show that masking must respect reasoning boundaries.** We perform an ablation where we randomly mask 0%, 40%, and 80% of reasoning tokens, instead of using DePO’s structured mask. On LogP, MR, and QED, these variants underperform full DePO:
>
> | LogP | SuccessRate $\uparrow$ | Sim $\uparrow$ | SuccessRate $\times$ Sim $\uparrow$ |
> | - | - | - | - |
> | **DePO (Ours)** | **0.415** | **0.715** | **0.297** |
> | DePO (Random Mask 80%) | 0.316 | 0.703 | 0.222 |
> | DePO (Random Mask 40%) | 0.233 | 0.747 | 0.174 |
> | DePO (No Mask) | 0.060 | 0.900 | 0.054 |
>
> | MR | SuccessRate $\uparrow$ | Sim $\uparrow$ | SuccessRate $\times$ Sim $\uparrow$ |
> | - | - | - | - |
> | **DePO (Ours)** | **0.399** | **0.736** | **0.294** |
> | DePO (Random Mask 80%) | 0.332 | 0.757 | 0.251 |
> | DePO (Random Mask 40%) | 0.226 | 0.782 | 0.177 |
> | DePO (No Mask) | 0.067 | 0.888 | 0.059 |
>
> | QED | SuccessRate $\uparrow$ | Sim $\uparrow$ | SuccessRate $\times$ Sim $\uparrow$ |
> | - | - | - | - |
> | **DePO (Ours)** | **0.312** | **0.756** | **0.236** |
> | DePO (Random Mask 80%) | 0.184 | 0.759 | 0.140 |
> | DePO (Random Mask 40%) | 0.127 | 0.771 | 0.098 |
> | DePO (No Mask) | 0.041 | 0.886 | 0.036 |
>
> **(3) LLM-as-judge evaluation quantifies improvements in reasoning quality.** We introduce a quantitative evaluation of reasoning quality using an LLM-as-judge protocol calibrated to expert chemists’ assessments [1]. For TOMG-Bench LogP optimization, we sample 50 trajectories per method and score the reasoning quality of each explanation. The results are shown below:
>
> | Method | Score $\uparrow$ |
> | - | - |
> | Baseline | 4.16 $\pm$ 1.32 |
> | GRPO | 3.60 $\pm$ 1.39 |
> | DePO (No mask) | 3.54 $\pm$ 1.15 |
> | DePO (Random mask 40%) | 4.18 $\pm$ 1.61 |
> | DePO (Random mask 80%) | 4.24 $\pm$ 1.70 |
> | **DePO (Ours)** | **4.32 $\pm$ 1.63** |
>
> Full DePO consistently attains the highest reasoning score, while GRPO and DePO (No Mask) lag behind. This shows that **gradient masking yields trajectories that are judged to be more coherent and chemically plausible**.

---

> ### Author Response · Authors · 2025-11-27
> **Response to Reviewer knP9 (part 5/7)**
>
> **(4) Training-log analysis shows that masking improves convergence and optimization stability.** We further analyze the training dynamics by tracking the average property improvement $\Delta \text{Prop}$, the higher the value, the better the performance, over iterations for DePO and DePO (No Mask), shown in Fig. 11.
> - For DePO, the smoothed $\Delta \text{Prop}$ curve steadily increases from around $0.05$ at the beginning to roughly $0.3$-$0.4$ at the end.
> - For DePO (No Mask), the smoothed curve remains in a narrow band around $0.08$–$0.12$ with no clear upward trend.
>
> Thus, **DePO with masking exhibits clear, monotonic improvement in the optimized property, whereas the no-mask variant saturates early and fluctuates around a low level**. This addresses the concern about convergence and optimization stability and shows that masking helps the RL signal interact constructively with the supervised objective rather than causing noisy, ineffective updates.
>
> In summary, our new experiments show that (i) arbitrary masking harms performance, (ii) DePO with structured masking is rated higher by an LLM-as-judge aligned with expert chemists, and (iii) masking stabilizes training and yields stronger property improvements.
>
> In the revision, we will (a) add the random-masking ablation tables for LogP, MR, and QED; (b) include the quantitative reasoning-quality table; (c) present the $\Delta \text{Prop}$ training-log plot; and (d) expand Sec. 4.2 to clarify the motivation and role of gradient masking in DePO.
>
> Reference:
> [1] Reasoning-Enhanced Large Language Models for Molecular Property Prediction. In arXiv, 2025.
>
> ### W5: How to create the demonstration dataset is not discussed.
> > how to create the demonstration dataset is not discussed.
>
> **Reply:** Thank you for this question. In brief, **the demonstration dataset is automatically derived from the ground truth optimized molecules already provided in the training splits of TOMG-Bench and MuMOInstruct, as described in Appendix E.6**, and we will make both the construction process and the meaning of “demonstration” more explicit in the main text.
>
> **(1) Demonstrations are constructed from benchmark ground truth optimized molecules, without extra human annotation.** For both TOMG-Bench and MuMOInstruct, the benchmarks already provide or imply optimized molecules that satisfy property and similarity constraints. We use these as demonstrations. Concretely, TOMG-Bench provides ground truth optimized molecules generated by RDKit-based modification rules, which are designed to improve a target property while keeping $\text{Similarity}(m^{*}, m_{0}) \ge \delta$. MuMOInstruct uses multi-property filtering with structural similarity thresholds to select such optimized molecules. All candidate molecules are validated for chemical correctness and required property improvements, so each pair $(q, a)$ corresponds to a feasible and high-quality optimization example.
>
> **(2) By “demonstration” we mean an input-output optimization example only.** In DePO, a demonstration refers to a pair consisting of a natural language query and its corresponding optimized molecule, for example:
>
> > Query: “Please modify the molecule `CCCC(C(=O)OC)C(O)C1CCCCC1` to increase its LogP value.”
> > Demonstration: `O=C(OC(CCCCO)CCCCS)c1ccccc1`
>
> In this case, the demonstration improves the LogP by $0.938$ compared to the original ones while preserving structural similarity (Sim$(m_0, m^*)=0.853$). We emphasize that demonstrations **contain only the final optimized molecule**, without step-by-step human annotations or chain-of-thoughts. The model learns to connect the instruction and the final optimized structure; all intermediate reasoning is generated and optimized by the policy itself under the DePO objective. This clarifies that “demonstration” in our work means benchmark-derived input–output examples, not manually curated reasoning traces.
>
> In summary, the demonstration dataset is automatically derived from the ground truth optimized molecules in the benchmark training splits and consists of instruction-optimized molecule pairs $(q, a)$, without any additional human annotation or reasoning labels. Demonstrations serve as limited supervision to anchor DePO’s exploration toward valid, high-quality solutions.
>
> In the revised manuscript, we will (i) move a concise description of this construction from Appendix E.6 into Sec. 3, (ii) explicitly define what we mean by “demonstration” (an instruction plus final optimized molecule, without human reasoning), and (iii) clearly state that all demonstrations come from benchmark training data generated by RDKit-based rules or multi-property filtering with similarity thresholds.

---

> ### Author Response · Authors · 2025-11-27
> **Response to Reviewer knP9 (part 6/7)**
>
> ### Q1: Clarification of the regularization term
> > The design and formulation of the regularization component are not clearly explained.
>
> **Reply:** Thank you for this question. In brief, **the regularization term in our objective is the standard PPO-style KL penalty inherited from GRPO for training stability**. We will make this connection clearer in the revised manuscript.
>
> **(1) The KL regularizer is inherited from GRPO and used only to stabilize policy updates.** In GRPO, the policy is optimized with a clipped PPO-style objective [1] that includes a KL penalty $\gamma\mathrm{KL}\bigl(\pi_\theta || \pi_{\theta_{\mathrm{old}}}\bigr)$, which controls the step size between consecutive policy iterates. This is the standard PPO mechanism for maintaining a trust region and preventing overly large policy updates; it serves **policy stability**, not task-specific exploration. In our method, we **reuse exactly the same KL term (Eqn. 3) and the same coefficient $\gamma$ as in GRPO**, without any extra tuning, so that the underlying optimization dynamics match GRPO. This choice ensures that any observed differences between GRPO and DePO are due to the demonstration-guided component, not to changes in the regularizer.
>
> **(2) We keep the KL coefficient fixed to the GRPO setting to isolate DePO’s contribution and avoid extra hyperparameter freedom.** Since $\gamma$ is inherited directly from GRPO, we do not treat it as a DePO-specific hyperparameter. As in prior PPO and GRPO work, $\gamma$ is chosen once to achieve stable updates and then held fixed. Changing $\gamma$ would introduce an additional degree of freedom that could confound the comparison by making GRPO and DePO operate under different trust region sizes. By fixing $\gamma$ to the original GRPO value and only varying $\beta$, we isolate the effect of the demonstration component and ensure that DePO’s improvements are not due to tuning the KL regularizer.
>
> In summary, the regularization component in our objective is simply the standard PPO-style KL penalty inherited from GRPO and used only for policy stability, while DePO’s novelty lies in the additional demonstration-guided term weighted by $\beta$. The role of $\gamma$ is unchanged from GRPO and is not responsible for the performance gains, which instead arise from the principled demonstration design and are robust across $\beta$ settings.
>
> In the revised manuscript, we will (i) explicitly separate the GRPO-inherited KL regularizer from the DePO-specific demonstration term in the objective, (ii) clearly state that $\gamma$ is fixed to the original GRPO value and serves only to stabilize policy updates, and (iii) cross-reference Fig. 5 to highlight that $\beta$ is the only new hyperparameter introduced by DePO and that our results are robust to its choice.
>
> Reference:
> [1] Proximal policy optimization algorithms. In arXiv, 2017.
>
> ### Q3. Reliability of the DePO-GRPO comparison in Figure 7
> > In Figure 7, there is a large discrepancy between DePO and GRPO. The result appears somewhat unreliable, as GRPO seems to lack a valid expression. Were both methods fine-tuned on the same base model? Without providing the source code, it's hard to believe such difference considering on overall similarity in objective function.
>
> **Reply:** Thank you for this careful question. **We would like to clarify that DePO and GRPO are fine-tuned from the same base model under the same training setup and that the difference in Fig. 7 is a real and systematically measured effect, not a consequence of mismatched models.** We also provide additional case studies in Appendix K.1, where GRPO cannot provide valid modifications. Below we explain (i) the shared implementation details and (ii) how our additional LLM-as-judge experiment quantitatively confirms the qualitative observations.
>
> **(1) Both DePO and GRPO are trained on exactly the same base model and training pipeline, with code provided.** For fairness, we initialize both methods from the same pretrained backbone, with the same tokenizer, the same molecular generation format, and the same training data splits. The GRPO implementation in our code follows the original GRPO formulation, including the group relative reward and the PPO-style KL regularization term with the same coefficient $\gamma$. DePO uses the identical GRPO implementation and hyperparameters as its RL backbone and simply adds the demonstration-guided term weighted by $\beta$. The complete training scripts and model implementations for both GRPO and DePO are already included in Appendix B of the original submission.

---

> ### Author Response · Authors · 2025-11-27
> **Response to Reviewer knP9 (part 7/7)**
>
> **(2) The additional LLM-as-judge experiment confirms that DePO’s reasoning quality advantage is systematic.** To avoid relying only on visual inspection, we add a quantitative evaluation of reasoning quality using an LLM as a judge protocol calibrated to expert chemists. On TOMG Bench LogP, we sample 50 trajectories per method and score the step-by-step reasoning for the baseline, GRPO, DePO without masking, random masking variants, and full DePO. DePO with its full design achieves the highest average reasoning score ($4.32 \pm 1.63$), while GRPO and DePO (no mask) obtain clearly lower scores, and random mask variants lie in between. **We kindly refer the reviewer to the responses in W3&W4&Q2 for more details.** This experiment uses the same base model and training configuration for GRPO as in Fig. 7, and yet the gap in reasoning scores remains consistent.
>
> In summary, the discrepancy between DePO and GRPO in Fig. 7 arises from their different objectives, not from different base models or mismatched implementations. Both methods are trained on the same backbone, with the same code base and hyperparameters, and the qualitative difference in reasoning traces is supported by a quantitative LLM-as-judge evaluation.
>
> In the revision, we will (i) explicitly state that DePO and GRPO share the same base model and training configuration, (ii) include the LLM-as-judge table to quantitatively support the reasoning quality gap, and (iv) clarify in the text that the qualitative examples in Fig. 7 are representative of the broader evaluation set rather than isolated outliers.
>
> **We thank Reviewer knP9 again** for the careful review and for considering our rebuttal. **If our responses have addressed your concerns, we would be very grateful if you could reconsider your overall rating.** Any further comments or questions you may have are welcome!

---

### Official Review · Reviewer_2riE · 2025-11-05

**Soundness:** 3
**Presentation:** 4
**Contribution:** 2
**Rating:** 4
**Confidence:** 4

**Summary:**

The paper addresses a key limitation of LLMs in scientific reasoning tasks, especially molecular optimization, designing molecules with better drug-like properties while maintaining structural similarity. Existing approaches like: SFT need reasoning traces, which are absent in molecule datasets, RLVR (Reinforcement Learning with Verifiable Rewards, e.g., GRPO) performs poorly due to sparse rewards and unguided exploration. Thus, the paper introduces DePO (Demonstration-guided Policy Optimization), a method that combines the benefits of RL and supervision using reference molecules as demonstrations to guide the optimization process.

The core idea is to guide the policy optimization process with demonstration molecules that serve as structural anchors, balancing exploration (via RL) and exploitation (via supervision). The task evaluated is molecular optimization i.e., given an input molecule (in SMILES form) and instruction (e.g., “make it more drug-like”), an optimized molecule needs to be generated. The training objective (Eq. 4) combines three components: RL term (Encouraging Exploration): Similar to GRPO, updates policy toward higher rewards, supervised term (Guiding Exploration): Encourages the model to generate solutions similar to demonstration molecules, regularization term: KL divergence to keep policy updates stable. Gradient masking ensures that only final answers (not reasoning tokens) are supervised, preserving reasoning ability. Two main rewards are proposed (a) structural similarity reward (r_struct): Tanimoto coefficient between input and generated molecule; (b) property reward (r_prop): Binary reward indicating whether target property (QED, LogP, MR) improves.

The paper evaluates DePO on two benchmarks: TOMG-Bench and MuMOInstruct. The metric used is the product of Success Rate (achieving property objective) and Structural Similarity (Tanimoto score).

**Strengths:**

- The paper is very well-written and easy to understand. I like how the problem is first formulated and clearly defined in Section 2, followed by a discussion of specific limitations in Section 3. The proposed method is then presented clearly and succinctly in Section 4, making it easy for readers to follow and grasp the details.

- DePO is an elegant fusion of RL and supervised signals, guiding exploration with domain demonstrations while maintaining reasoning depth. It demonstrates constraint search to chemically meaningful regions, overcoming limitations of GRPO.

- The paper presents strong empirical results. DePO outperforms SFT and GRPO on multiple benchmarks (TOMG-Bench, MuMOInstruct) by up to 13%.

**Weaknesses:**

- In the DePO scheme, the LLM output is first parsed into reasoning tokens and the generated final answer. Then, the generated final answer is replaced with the gold-standard final answer. This way, the reasoning tokens are preserved, and gradient masking for the intermediate reasoning steps excludes these tokens from parameter updates during optimization. The authors claim that this approach prevents the LLM from learning potentially erroneous reasoning patterns, but it is not clear why that is the case. There is no constraint imposed on the reasoning tokens that prevents the model from generating erroneous reasoning patterns. In fact, training only occurs on tokens excluding the reasoning tokens. The authors further state that DePO "constrains its exploration to chemically valid and promising regions of the solution space," but it is not evident why this is true. Moreover, what is the rationale behind replacing the generated answer with the gold-standard answer? The reward is anyway calculated based on the generated answer. The overall design seems to offer only incremental improvements over GRPO.

- For the reward assessing the target property, a binary reward value of 1 is assigned if the generated molecule achieves a favorable change in the target property relative to the original molecule. This means that even a minuscule positive change results in a reward of 1. Such a design is highly susceptible to reward hacking, where the model may settle after making only a small improvement and fail to pursue further optimization, since it has already received the maximum reward of 1.

- Why is the product of Success Rate (achieving the property objective) and Structural Similarity (Tanimoto score) considered a meaningful metric? Wouldn’t this metric be biased if one component is significantly high while the other is only moderate? From Section 2.1, it seems more reasonable to report these as separate metrics: (a) drug-likeness, conditioned on structural similarity being within a specified threshold (0 otherwise), and (b) structural similarity itself.

- The paper only compares against different fine-tuning paradigms, namely, Baseline, SFT, GRPO, and GRPO (SFT init). However, it does not include comparisons with existing state-of-the-art methods for molecular optimization, nor with any foundation models.

- Figure 2 is difficult to interpret, and the corresponding observations are not clearly conveyed. For instance, Observation 3.1 ("Models trained with GRPO exhibit a conservative bias, generating molecules nearly identical to the input, as shown in Fig. 2 (Left)") is not directly supported or easily interpretable from the figure. I suggest modifying it to a different type of plot that more clearly illustrates the stated observations and providing a more descriptive caption to enhance interpretability.

**Questions:**

See weaknesses.

---

> ### Author Response · Authors · 2025-11-27
> **Response to Reviewer 2riE (part 1/8)**
>
> **We thank Reviewer 2riE for the thoughtful and detailed feedback.** In this rebuttal, we try our best to address all of your concerns regarding (i) gradient masking, answer replacement, and demonstration guidance in DePO; (ii) the binary property reward and reward-hacking risks; (iii) the SuccessRate $\times$ Similarity metric; (iv) comparisons with molecular optimization baselines and foundation models; and (v) the clarity of Fig. 2. Specifically:
> - First, we clarify how gradient masking and answer replacement interact with the RL objective and show that improper or no masking significantly harms optimization performance.
> - Second, we analyze the binary property reward together with similarity, compare against a continuous variant, and show that DePO still pursues substantial property gains.
> - Third, we justify the Success Rate $\times$ Similarity metric as the official TOMG-Bench objective, report each component separately, and add the requested conditional analysis.
> - Fourth, we expand our evaluation to additional molecular optimizers and foundation models, showing that DePO remains competitive or superior.
> - Finally, we redesign Fig. 2 using violin plots and a clearer caption so that GRPO’s conservative behavior and DePO’s improved exploration are more visible.
>
>
> Point-by-point responses are provided below, together with the corresponding planned revisions.
>
> ### W1.1: Clarification on gradient masking and thinking token updates
> > In the DePO scheme, the LLM output is first parsed into reasoning tokens and the generated final answer. Then, the generated final answer is replaced with the gold-standard final answer. This way, the reasoning tokens are preserved, and gradient masking for the intermediate reasoning steps excludes these tokens from parameter updates during optimization. The authors claim that this approach prevents the LLM from learning potentially erroneous reasoning patterns, but it is not clear why that is the case. There is no constraint imposed on the reasoning tokens that prevents the model from generating erroneous reasoning patterns. In fact, training only occurs on tokens excluding the reasoning tokens.
>
> **Reply:** Thank you for this insightful comment. In brief, **DePO’s gradient-masking mechanism prevents potentially incorrect supervision from the model-generated thinking tokens, while the RL objective continues to optimize all thinking tokens through reward-driven optimization.** Because demonstrations contain only the final molecule, masking is essential to avoid forcing the model to align potentially incorrect reasoning with correct answers. **Tab. 3 and additional experiments on the masking strategy confirm that removing the mask significantly degrades optimization performance.**
>
> **(1) All thinking tokens remain trainable under the RL objective (the “Exploration term” in Eqn. 4), while gradient masking is applied only to the “Guidance term” $\log \pi_\theta(a_i \mid q, t_i)$.** The RL term continues to propagate gradients to every thinking token through the advantage values $\hat{A}_{i,k}$, which is computed from the final reward $r_i$, so reasoning patterns leading to poor molecules receive negative updates, while successful ones are reinforced. We also refine Fig. 4 to better demonstrate the underlying mechanism.
>
> **(2) Masking is necessary because demonstrations contain only the final optimized molecule and no thinking steps, so the demonstration and the model-generated reasoning are inherently misaligned for full-sequence supervision.** Without masking, the supervised loss would be applied to the entire sequence, implicitly treating the self-generated thinking tokens as if they were the correct reasoning that produced the demonstration molecule. This misalignment makes the supervision inconsistent: the model is effectively encouraged to preserve reasoning patterns where incorrect reasoning leads to the correct molecule. By restricting the demonstration gradient to the final-answer tokens, the mask avoids this potentially misaligned gradient signal, while the thinking tokens are optimized solely through the task-aligned reward signal.

---

> ### Author Response · Authors · 2025-11-27
> **Response to Reviewer 2riE (part 2/8)**
>
> **(3) Empirical ablations validate the necessity of the masking mechanism.** Tab. 3 shows that removing the gradient mask results in a substantial drop in optimization performance across all TOMG-Bench tasks. This degradation reflects the reasoning-answer misalignment that masking is designed to prevent. In addition to the ablation already included in Tab. 3, we evaluate DePO with 0\%, 40\%, and 80\% random masking of the reasoning tokens on the LogP, MR, and QED tasks. As shown in the following table, these masking variants significantly **underperform** DePO, indicating that arbitrary masking disrupts the consistency between the reasoning prefix and the supervised answer tokens.
>
> | LogP | SuccessRate $\uparrow$ | Sim $\uparrow$ | SuccessRate $\times$ Sim $\uparrow$ |
> | - | - | - | - |
> | DePO (Ours) | **0.415** | **0.715** | **0.297** |
> | DePO (Random Mask 80%) | 0.316 | 0.703  | 0.222 |
> | DePO (Random Mask 40%) | 0.233 | 0.747 | 0.174 |
> | DePO (No Mask) | 0.060 | 0.900 | 0.054 |
>
> | MR | SuccessRate $\uparrow$ | Sim $\uparrow$ | SuccessRate $\times$ Sim $\uparrow$ |
> |-|-|-|-|
> | DePO (Ours) | **0.399** | **0.736** | **0.293** |
> | DePO (Random Mask 80%) | 0.332 | 0.757 | 0.251 |
> | DePO (Random Mask 40%) | 0.226 | 0.782 | 0.177 |
> | DePO (No Mask) | 0.067 | 0.888 | 0.059 |
>
> | QED | SuccessRate $\uparrow$ | Sim $\uparrow$ | SuccessRate $\times$ Sim $\uparrow$ |
> |-|-|-|-|
> | DePO (Ours) | **0.312** | 0.756 | **0.236** |
> | DePO (Random Mask 80%) | 0.184 | 0.759 | 0.140 |
> | DePO (Random Mask 40%) | 0.127 | 0.771 | 0.098 |
> | DePO (No Mask) | 0.041 | 0.886 | 0.036 |
>
> **In addition, we examine training dynamics to understand how masking influences optimization.** Fig. 9 compares reward trajectories between GRPO and DePO, while Fig. 11 contrasts full DePO with its no-mask variant. These curves consistently show that DePO maintains more stable and steadily improving reward signals. Further qualitative examples in Appendix J.1 show that GRPO frequently drift into chemically implausible or reward-inconsistent trajectories. Notably, DePO with gradient masking consistently achieves **higher** reward and **better** reasoning behavior, confirming that the masking mechanism stabilizes training while preserving the benefits of demonstration guidance.
> We also evaluate reasoning quality using an LLM-as-judge protocol calibrated to expert chemists’ assessments [1]. For TOMG-Bench LogP, we score 50 trajectories per method:
>
> | Method | Score $\uparrow$ |
> | - | - |
> | Baseline | 4.16 $\pm$ 1.32 |
> | GRPO | 3.60 $\pm$ 1.39 |
> | DePO (No mask) | 3.54 $\pm$ 1.15 |
> | DePO (Random mask 40%) | 4.18 $\pm$ 1.61 |
> | DePO (Random mask 80%) | 4.24 $\pm$ 1.70 |
> | **DePO (Ours)** | **4.32 $\pm$ 1.63** |
>
> Full DePO achieves the highest score of reasoning quality, while GRPO and DePO with different masking strategies consistently lag behind. This confirms that **gradient masking yields more coherent, chemically plausible, and reward-consistent reasoning trajectories** and is therefore an essential component of the method.
>
> In summary, gradient masking is a principled and essential component of DePO that protects the reasoning process from inconsistent supervision while allowing the RL objective to optimize reasoning in a reward-driven manner.
>
> We will revise the manuscript to emphasize this rationale, clarify why full-sequence supervision is incompatible with demonstrations lacking reasoning trajectories, add the additional experiments to the appendix, and highlight the supporting evidence from Tab. 3.
>
> Reference:
> [1] Reasoning-Enhanced Large Language Models for Molecular Property Prediction. In arXiv, 2025.

---

> ### Author Response · Authors · 2025-11-27
> **Response to Reviewer 2riE (part 3/8)**
>
> ### W1.2: Explanation of how DePO constrains exploration to valid chemical regions
> > The authors further state that DePO "constrains its exploration to chemically valid and promising regions of the solution space," but it is not evident why this is true.
>
> **Reply:** Thank you for raising this point. **DePO constrains exploration because its supervised guidance term is built from RDKit-validated, property-improving molecules that serve as reliable anchors in chemical space, while the RL component encourages exploration for further improvements.** This grounding keeps the search within chemically valid and high-reward regions and prevents the exploration collapse observed in GRPO. As shown in Fig. 8, DePO achieves much larger and more stable property gains than GRPO and even surpasses the average improvement of the demonstrations.
>
> **(1) Demonstration guidance anchors the policy around verified and chemically meaningful molecules.** Each demonstration molecule satisfies strict RDKit validity, property improvement, and similarity constraints (Appendix E.6 in the original submission). By maximizing $\log \pi_\theta(a_i \mid q, t_i)$ toward these curated molecules, **DePO biases its exploration toward regions of chemical space that are structurally feasible and directly relevant for the target properties.** This keeps policy updates aligned with chemically meaningful scaffolds rather than drifting into invalid or unproductive regions. Additionally, we sample 100 subsamples from the validation set from the trained model, shown in Fig. 8. Notably, DePO attains a much higher mean $\Delta \text{MR}$ than GRPO and even exceeds the mean gain of the demonstration set, indicating that the policy explores around and beyond these high-quality demonstrations instead of collapsing to trivial or degenerate solutions.
>
> **(2) Unguided RL exploration in GRPO reveals the need for this constraint.** Molecular optimization involves a combinatorially large search space, and vanilla GRPO lacks any mechanism to indicate which parts of this space are chemically promising. As shown in Fig. 2 (left), GRPO often collapses into generating molecules nearly identical to the input, achieving limited property improvement and exploiting the similarity constraint without productive exploration. Fig. 9 further illustrates that GRPO’s success rate and property gain curves remain low throughout training, while DePO’s curves steadily increase and stabilize at much higher values. This contrast indicates that DePO’s demonstration guidance provides a strong directional signal that steers exploration toward regions where valid, property-improving molecules are dense, rather than allowing the policy to stagnate near the starting structures.
>
> **(3) Empirical evidence confirms that DePO’s guided exploration improves both validity and diversity.** Appendix E.2 has reported that DePO generates an average of 34 unique valid molecules, compared with only 6 from GRPO under the same setup, showing that DePO explores a broader set of valid candidates. Fig. 10 further visualizes the distributions of $\Delta\text{LogP}$, $\Delta\text{MR}$, and $\Delta\text{QED}$: DePO shifts the density toward larger property gains compared with GRPO and the baseline, while concentrating most samples in a property-improving range instead of near-zero changes.
>
> In summary, the demonstration-guided supervision is the mechanism that constrains exploration to chemically meaningful regions, while the RL component maintains reward-driven diversity around these anchors.
>
> We will revise the manuscript to more clearly explain how these constraints arise from the demonstrations.

---

> ### Author Response · Authors · 2025-11-27
> **Response to Reviewer 2riE (part 4/8)**
>
> ### W1.3: Motivation for answer replacement within the supervised guidance term
> > Moreover, what is the rationale behind replacing the generated answer with the gold-standard answer? The reward is anyway calculated based on the generated answer.
>
> **Reply:** Thank you for raising this point. **Answer replacement is introduced so that the supervised guidance term can evaluate both the model-generated answer and the demonstrated (ground-truth) answer under the same contextual information, that is, conditioned on the same reasoning prefix $(q, t_i)$.** This alignment makes the answer logits comparable and stabilizes the supervised regularizer, while the RL reward remains computed on the model’s own generated molecule.
>
> **(1) The supervised loss requires computing answer log-probabilities under a shared context $(q, t_i)$.** In DePO, the policy is autoregressive, so the probability of an answer sequence depends on its preceding tokens. For the demonstrated molecule $a_i^\ast$ and the model-generated molecule $\hat{a_i}$, we would like to compare or regularize quantities of the form $\log \pi_\theta(a_i^\ast \mid q, t_i)$ and $\log \pi_\theta(\hat{a_i} \mid q, t_i)$, that is, both answers conditioned on the **same** reasoning context $(q, t_i)$. If we keep $\hat{a_i}$ in the sequence and directly apply the supervised loss, the model effectively evaluates $\log \pi_\theta(a_i^\ast \mid q, t_i, \hat{a_i})$, which is conditioned on a different and often invalid final token. This mismatch makes the logits underlying the supervised term inconsistent and can lead to unstable gradients.
>
> **(2) Replacing the generated answer with the demonstrated one ensures that both answers are evaluated under the same context.** By substituting $\hat{a_i}$ with $a_i^\ast$ only in the supervised branch, DePO ensures that the answer logits used in the demonstration-guided loss correspond to $\log \pi_\theta(a_i^\ast \mid q, t_i)$ and, when needed, $\log \pi_\theta(\hat{a_i} \mid q, t_i)$, both conditioned on the same reasoning prefix. This alignment keeps the supervised signal well-defined and comparable, rather than mixing probabilities computed under incompatible contexts.
>
> **(3) Importantly, this replacement does not alter how the RL reward is computed.** In the exploration term, the reward is still based on the model’s own generated molecule $\hat{a}_i$, so policy improvement remains fully reward-driven. The replacement only affects the supervised guidance term and is used to make the answer-level logits comparable and stable, not to shortcut the optimization.
>
> In summary, answer replacement is motivated by the need to evaluate gold and generated answers under the same reasoning context $(q, t_i)$, so that their log-probabilities are aligned and the supervised regularizer is well behaved, while the RL objective continues to operate on the model’s actual generations.
>
> We will clarify this motivation more explicitly in the revised manuscript and connect it to the formulation in Eqn. 4.

---

> ### Author Response · Authors · 2025-11-27
> **Response to Reviewer 2riE (part 5/8)**
>
> ### W1.4: Clarification on the scope and significance of DePO’s improvements over GRPO
> > The overall design seems to offer only incremental improvements over GRPO.
>
> **Reply:** Thank you for this thoughtful comment. In brief, **DePO is motivated by and explicitly addresses the fundamental failure modes of GRPO identified in our analysis, rather than providing an incremental modification.** GRPO lacks reliable priors and valid chain-of-thought guidance in this setting, which leads to collapsed exploration and degraded reasoning. DePO introduces a principled mechanism that corrects these deficiencies and delivers substantial empirical improvements.
>
> **(1) GRPO fails systematically in this domain due to the absence of valid reasoning supervision and the difficulty of exploring chemical space.** As shown in Sec. 3 and Fig. 2, GRPO trained from an SFT-initialized model (which has only question-answer pairs and no reasoning traces) is unable to recover multi-step reasoning ability (Obs. 3.2). GRPO also exhibits overly conservative exploration, repeatedly generating molecules almost identical to the input (Obs. 3.1), a consequence of the sparse reward and the similarity constraint inherent to TOMG-Bench. These issues reflect fundamental limitations of GRPO in settings where neither reliable priors nor curated reasoning trajectories are available.
>
> **(2) DePO is specifically designed to overcome these limitations through a demonstration-guided term and reward-driven reasoning optimization.** The demonstration term anchors the policy toward RDKit-validated, property-improving molecules while maintaining exploration through the RL update (Eqn. 4). At the same time, the RL advantage $\hat{A}_{i,k}$ continues to optimize all reasoning tokens based on reward, allowing DePO to explore chemically valid regions without collapsing into trivial edits. This complementary design is necessary to address the failure modes identified in Fig. 2, rather than a small addition to GRPO.
>
> **(3) The resulting improvements are substantial, consistent, and qualitatively distinct from GRPO.** Across all TOMG-Bench tasks, DePO achieves significant gains (up to +13 percent over GRPO shown in Tab. 1). The reasoning trajectories produced by DePO are also different: Fig. 7, additional case studies in Appendix J.1, and the LLM-as-Judge experiment show that DePO produces better optimization paths compared to GRPO.
>
> In summary, DePO represents a necessary and principled improvement to GRPO, directly targeting its documented limitations in exploration and reasoning, and the empirical results confirm that the improvements are far from incremental.
>
> We will revise the manuscript to make this connection more explicit and highlight how the design of DePO stems directly from the observed weaknesses of GRPO.
>
> ### W2: Clarification on binary property reward formulation and mitigation of reward hacking risks
> > For the reward assessing the target property, a binary reward value of 1 is assigned if the generated molecule achieves a favorable change in the target property relative to the original molecule. This means that even a minuscule positive change results in a reward of 1. Such a design is highly susceptible to reward hacking, where the model may settle after making only a small improvement and fail to pursue further optimization, since it has already received the maximum reward of 1.
>
> **Reply:** Thank you for this thoughtful comment. We would like to clarify that the binary property reward is a deliberate design choice in DePO’s framework: the method **supports both continuous and binary property rewards**, but we ultimately adopt the binary version because it yields more stable training and better optimization performance in our setting and empirically **does not lead to reward hacking** when combined with the similarity term and group-relative normalization.
>
> **(1) The final reward is not purely binary and cannot be maximized by trivial edits.** In DePO, the final reward is $r_i = r_\mathrm{prop}(m_i,m_0) + r_\mathrm{sim}(m_i,m_0)$, where $r_\mathrm{prop}\in\{0,1\}$ indicates whether the target property improves, and $r_\mathrm{sim}$ is a continuous similarity score to the initial molecule $m_0$. Thus, a molecule that achieves only a tiny positive change but severely disrupts similarity will obtain a **lower total reward** than one that both improves the property and remains close to $m_0$. Moreover, GRPO uses group-relative advantages $\hat{A}_{i,k}$, so trajectories with only marginal gains do not receive a larger advantage if other samples in the same group achieve better overall reward.

---

> ### Author Response · Authors · 2025-11-27
> **Response to Reviewer 2riE (part 6/8)**
>
> **(2) Continuous property rewards are supported but empirically underperform the binary variant.** We experimented with a continuous property reward that scales with the magnitude of the property improvement. As reported in Tab. 7, this variant consistently **underperforms** the binary formulation. We attribute this to: (i) **Scale sensitivity**, which inflates variance and destabilizes policy updates, and (ii) **credit-assignment dilution**, where large but rare improvements dominate gradient estimates while many small gains contribute noisy signals that do not translate into better molecules. In contrast, the binary reward provides a clean, low-variance success signal, and the GRPO group-relative normalization recovers a meaningful ranking among trajectories.
>
> **(3) Empirical analysis shows that DePO continues to pursue large improvements rather than settling at the threshold.** To directly check for reward-hacking behavior, we analyze the distribution of property improvement among successful samples (those with $r_\mathrm{prop}=1)$. As shown in Fig. 10, **DePO’s distributions of $\Delta\text{LogP}$, $\Delta\text{MR}$, and $\Delta\text{QED}$ are consistently shifted to the right with heavier tails compared to GRPO and the baseline.** Across all three properties, the mode of GRPO’s density lies just above zero, indicating that GRPO often exploits minimal improvements. In contrast, DePO produces broader, more diverse distributions with substantially larger mean and median gains, demonstrating that the policy continues to optimize beyond the binary threshold rather than stopping once $r_\mathrm{prop}$ is achieved. Additionally,  Fig. 9 further illustrates that GRPO’s success rate and property gain curves remain low throughout training, while DePO’s curves steadily increase and stabilize at much higher values.
>
> In summary, DePO’s full objective and training scheme mitigate this risk in practice: the similarity term and group-relative normalization prevent trivial threshold-level edits from dominating, demonstration guidance biases the policy toward non-trivial improvements, and our analyses (Tab. 7) show that DePO discovers better molecules compared to DePO with continuous rewards. We will clarify these points and the supporting evidence in the revised manuscript.
>
> ### W3: Metric Design
> > Why is the product of Success Rate (achieving the property objective) and Structural Similarity (Tanimoto score) considered a meaningful metric? Wouldn’t this metric be biased if one component is significantly high while the other is only moderate? From Section 2.1, it seems more reasonable to report these as separate metrics: (a) drug-likeness, conditioned on structural similarity being within a specified threshold (0 otherwise), and (b) structural similarity itself.
>
> **Reply:** Thank you for raising this point. The product of Success Rate and Structural Similarity follows the **official evaluation protocol of TOMG-Bench (Sec. 5.1)**, where the optimization objective in Eqn. 1 explicitly requires simultaneous property improvement and similarity preservation. The product metric provides a compact measure of this joint requirement: if either component is low, the combined score decreases proportionally.
>
> **(1) We agree that reporting the two components separately improves interpretability.** In the original submission, **we have reported Success Rate and Similarity as standalone metrics in Appendix G.1.** In the revision, we will move these per-component metrics into the main tables alongside their product.
>
> **(2) We report the suggested conditional metric.** There, we first enforce the molecular similarity constraint (similarity $\ge$ 0.40 based on [1,2]) and then analyze the report property gains (e.g., mean of $\Delta\text{LogP}$, $\Delta\text{MR}$, $\Delta\text{QED}$) for molecules that satisfy this constraint. DePO exhibits consistently larger conditional property improvements than GRPO under this setting, indicating that its gains persist when explicitly conditioning on structural similarity.
>
> | Method | mean $\Delta$ QED | mean $\Delta$ MR | mean $\Delta$ LogP | mean similarity (QED) | mean similarity (MR) | mean similarity (LogP) |
> |-|-|-|-|-|-|-|
> | Baseline | 0.070 | 9.959 | 0.813 | 0.647 | 0.655 | 0.660 |
> | GRPO | 0.061 | 5.001 | 0.734 | 0.740 | 0.735 | 0.731 |
> | **DePO** | **0.099** | **19.054** | **1.257** | **0.704** | 0.679 | 0.701 |
>
> Importantly, DePO improves both Success Rate and Similarity simultaneously on TOMG-Bench and MuMOInstruct (Tabs. 1-2). Thus, the higher product score is not obtained by trading off one component against the other but reflects a genuine improvement in balanced molecular optimization.
>
> References:
> [1] The use of 2D fingerprint methods to support the assessment of structural similarity in orphan drug legislation. In Journal of Cheminformatics, 2014.
> [2] Lo-Hi: Practical ML Drug Discovery Benchmark. In NeurIPS, 2023.

---

> ### Author Response · Authors · 2025-11-27
> **Response to Reviewer 2riE (part 7/8)**
>
> ### W4: Comparison with existing molecular optimization methods and foundation models
> > The paper only compares against different fine-tuning paradigms, namely, Baseline, SFT, GRPO, and GRPO (SFT init). However, it does not include comparisons with existing state-of-the-art methods for molecular optimization, nor with any foundation models.
>
> **Reply:** Thank you for this helpful suggestion. In brief, our response is that (i) we **expand the evaluation** beyond fine-tuning paradigms to **include state-of-the-art molecular optimization methods** from multiple algorithmic families, and (ii) **the expanded results show that DePO remains competitive or superior in performance.**
>
>  **(1) We expand our evaluation to cover state-of-the-art molecular optimizers and foundation models beyond fine-tuning paradigms.** Thank you for this helpful suggestion. In the revision, we extend our comparison to include state-of-the-art molecular optimization methods from four major algorithmic families, as well as molecular and general-purpose foundation models. Concretely, we add optimization methods (Graph-GA [1], REINVENT [2], MOLLEO [3]), molecular foundation models (Bio-T5-base [4], Mol-T5-large [5]), and general LLMs (GPT-4o-mini), in addition to the methods in Tab. 1.
>
> Specifically,
> - Graph-GA operates directly on molecular graphs using RDKit-defined crossover and mutation operations derived from graph-matching rules, supporting both atom-level and fragment-level modifications.
> - REINVENT adopts an RL-based approach to tune an RNN for SMILES generation toward the target property.
> - MOLLEO integrates a pretrained LLM as a proposal generator within the Graph-GA framework to guide crossover and mutation, improving over purely random operators.
>
> Following the original MOLLO setup, we run 100 evolutionary iterations and keep the remaining hyperparameters at their default values for the above Graph-GA, REINVENT, and MOLLEO.
>
> - For molecular foundation models, we include Mol-T5-large and Bio-T5-base, which leverage large-scale biochemical text and molecular data to improve SMILES understanding and generation.
> - For general-purpose LLMs, we evaluate GPT-4o-mini.
>
> All these foundation models are used in a zero-shot setting: we prompt them with a natural language description of the optimization task and ask them to directly output a candidate molecule, then compute property improvement and similarity under the TOMG-Bench evaluation protocol.
>
> **(2) DePO remains competitive or superior across Success Rate and Similarity.** As shown in the following tables, DePO consistently achieves higher or comparable joint scores than Graph-GA, REINVENT, MOLLO, Mol-T5-large, and Bio-T5-base, while maintaining strong structural similarity. GPT-4o-mini achieves strong performance as a powerful proprietary LLM, yet DePO remains competitive with these models despite using a much smaller open-source backbone and operating in a fine-tuned regime specifically tailored for TOMG-Bench.
>
> | LogP | **Success Rate $\uparrow$** | **Sim. $\uparrow$** | **Success Rate $\times$ Sim. $\uparrow$** |
> | - | - | - | - |
> | Graph GA | 0.509 | 0.125 | 0.064 |
> | REINVENT | 0.465 | 0.125 | 0.058 |
> | MOLLO | 0.510 | 0.103 | 0.053 |
> | GPT-4o-mini | 0.499 | 0.706 | 0.352 |
> | Mol-T5-large | 0.424 | 0.102 | 0.043 |
> | Bio-T5-base | 0.516 | 0.153 | 0.079 |
> | Baseline (Qwen 2.5-3B Instruct) | 0.268 | 0.627 | 0.168 |
> | SFT | 0.298 | 0.692 | 0.206 |
> | GRPO | 0.379 | 0.806 | 0.305 |
> | GRPO (SFT-Init) | 0.212 | 0.863 | 0.183 |
> | **DePO (Ours)** | **0.415** | **0.715** | **0.297** |
>
> | MR | **Success Rate $\uparrow$** | **Sim. $\uparrow$** | **Success Rate $\times$ Sim. $\uparrow$** |
> | - | - | - | - |
> | Graph GA | 0.509 | 0.122 | 0.062 |
> | REINVENT | 0.595 | 0.116 | 0.069 |
> | MOLLO | 0.509 | 0.127 | 0.065 |
> | GPT-4o-mini | 0.409 | 0.771 | 0.315 |
> | Mol-T5-large | 0.450 | 0.107 | 0.048 |
> | Bio-T5-base | 0.506 | 0.160 | 0.081 |
> | Baseline (Qwen 2.5-3B Instruct) | 0.252 | 0.685 | 0.173 |
> | SFT | 0.359 | 0.663 | 0.238 |
> | GRPO | 0.214 | 0.880 | 0.188 |
> | GRPO (SFT-Init) | 0.265 | 0.850 | 0.225 |
> | **DePO (Ours)** | **0.399** | **0.736** | **0.294** |
>
> | QED | **Success Rate $\uparrow$** | **Sim. $\uparrow$** | **Success Rate $\times$ Sim. $\uparrow$** |
> | - | - | - | - |
> | Graph GA | 0.493 | 0.140 | 0.069 |
> | REINVENT | 0.558 | 0.115 | 0.064 |
> | MOLLO | 0.496 | 0.122 | 0.061 |
> | GPT-4o-mini | 0.231 | 0.752 | 0.174 |
> | Mol-T5-large | 0.465 | 0.119 | 0.055 |
> | Bio-T5-base | 0.507 | 0.158 | 0.080 |
> | Baseline (Qwen 2.5-3B Instruct) | 0.188 | 0.693 | 0.130 |
> | SFT | 0.297 | 0.697 | 0.207 |
> | GRPO | 0.138 | 0.889 | 0.123 |
> | GRPO (SFT-Init) | 0.223 | 0.863 | 0.193 |
> | **DePO (Ours)** | **0.312** | **0.756** | **0.236** |
>
> Importantly, these expanded baselines confirm that DePO performs competitively across diverse categories of molecular optimization methods.
>
> In the revision, we will integrate these results into the paper and update our discussion to reflect this broader comparison.

---

> ### Author Response · Authors · 2025-11-27
> **Response to Reviewer 2riE (part 8/8)**
>
> References:
> [1] A graph-based genetic algorithm and generative model/Monte Carlo tree search for the exploration of chemical space. In Chemical Science, 2019.
> [2] Molecular de-novo design through deep reinforcement learning. In Journal of Cheminformatics, 2017.
> [3] Efficient Evolutionary Search Over Chemical Space with Large Language Models. In ICLR, 2025.
> [4] Biot5: Enriching cross-modal integration in biology with chemical knowledge and natural language associations. In EMNLP, 2023.
> [5] Translation between molecules and natural language. In EMNLP, 2022.
>
> ### W5: Improve the interpretability of Figure 2.
> > Figure 2 is difficult to interpret, and the corresponding observations are not clearly conveyed. For instance, Obs. 3.1 ("Models trained with GRPO exhibit a conservative bias, generating molecules nearly identical to the input, as shown in Fig. 2 (Left)") is not directly supported or easily interpretable from the figure. I suggest modifying it to a different type of plot that more clearly illustrates the stated observations and providing a more descriptive caption to enhance interpretability.
>
> **Reply:** Thank you for this suggestion. In the revision, we improve the figure design to make the stated observations visually explicit and easier to interpret.
>
> **We replace the original histograms with violin plots that compare distribution shapes across methods.** Violin plots provide a much clearer view of how each method distributes its outputs. The revised Fig. 2 now includes a similarity violin plot across methods (GRPO vs. DePO) and a Success Rate violin plot.
>
> This enhances the visualization by directly showing **GRPO’s similarity values are tightly concentrated near 1.0**, indicating that it frequently generates molecules nearly identical to the input. In contrast, **DePO maintains high similarity but exhibits a broader distribution**, reflecting less conservative exploration under the similarity constraint.
>
> In summary, the updated figure uses violin plots to highlight distributional differences across methods, making GRPO’s conservative behavior and DePO’s improved exploration more visible.
>
> **We thank Reviewer 2riE again** for the careful review and for considering our rebuttal. **If our responses have addressed your concerns, we would be very grateful if you could reconsider your overall rating.** Any further comments or questions you may have are welcome!

---

### Meta-Review · Area_Chair_7b6H · 2026-01-05

**Summary:**

This submission provides a novel way of combining demonstration and RLVR in Molecular Optimization to address the challenges of standard GRPO in this problem. Multiple reviewers aggree it provides strong empirical performance on multiple benchmarks.

The main concerns are as follows,
1. (2riE, knP9, h5YJ) Incremental novelty compared to GRPO (+ imitation learning)
2. (2riE, knP9, h5YJ) Confusion of the rationale behind "Answer Replacement" and "Gradient Masking" and lack of theoretical and detailed analysis
3. (2riE, knP9, h5YJ) Lack of discussion on deign choices including: binary reward, evaluation metric, hyperparameters
4. (2riE) Lack of baselines beyond fine-tuning paradigms
5. (knP9) Limited discussion of GPT-based molecular optimization studies.
6. (knP9) Subjective evaluation of reasoning quality
7. (h5YJ, Aw6o) Generalization to other scientific or reasoning domains, low-quality demonstration, stronger base models
8. (Aw6o) Lack of range of errors in addition to the mean

**Reviewer Concerns:**

The authors' rebuttal provided detailed discussions and additional experiments to all the concerns and questions.

Concern 2, 3, 5, 8 are well explained with additional ablations and experimental results.

Concern 1: the authors clearly explains the novelty from a standard GRPO. While it is in essence of hybrid GRPO and imitation learning, it provides a clever way of avoiding the shortcoming of SFT through gradient masking.

Concern 4: additional baselines are provided including traditional optimization methods, molecular foundation models, and general LLMs. It shows competitive performance across all baselines except GPT-4o-mini (the proposed method is finetuned on a 3B model)

Concern 6: authors addressed the concern by adding an LLM-as-a-judge evaluation of reasoning traces to provide a quantitative metric.

Concern 7: partially addressed. Authors provided additional experiments on the mathematical reasoning benchmark (MATH), however the improvement is marginal. Also provided ablations for low-quality demonstration, stronger base models which demonstrate the generalization in those settings.

**Reviewer Scores:**

2riE, initial 4, main concerns: 1, 2, 3, 4. Likely to raise the score to 6.

knP9, initial 4, main concerns 1, 2, 3, 5, 6. Most concerns are well addressed. Likely to raise the score to 6.

h5YJ, initial 6, main concerns: 1, 2, 3, 7. The new experiments in MATH is not sufficient to address the concern of generalization to other scientific or reasoning domains. And the concern of the lack of theoretical insights might not be addressed satisfiable. Other concerns are well addressed. Likely to maintain the score.

Aw6o, initial 4, main concerns: 7, 8. The concern of generalization to stronger base models and lack of error range are well addressed. Likely to raise the score to 6.

---

### Decision · Program_Chairs · 2026-01-26

Accept (Poster)